# Activation of the glutamatergic cingulate cortical-cortical connection facilitates pain in adult mice

Xu-Hui Li [1,2,3,7], Wantong Shi[1,3,7], Qi-Yu Chen [3,4], Shun Hao[3], Hui-Hui Miao[2,5], Zhuang Miao[2], Fang Xu[4], Guo-Qiang Bi [4] & Min Zhuo [1,2,3,6✉]

The brain consists of the left and right cerebral hemispheres and both are connected by callosal projections. Less is known about the basic mechanism of this cortical-cortical connection and its functional importance. Here we investigate the cortical-cortical connection between the bilateral anterior cingulate cortex (ACC) by using the classic electro-physiological and optogenetic approach. We find that there is a direct synaptic projection from one side ACC to the contralateral ACC. Glutamate is the major excitatory transmitter for bilateral ACC connection, including projections to pyramidal cells in superficial (II/III) and deep (V/VI) layers of the ACC. Both AMPA and kainate receptors contribute to synaptic transmission. Repetitive stimulation of the projection also evoked postsynaptic $Ca^{2+}$ influx in contralateral ACC pyramidal neurons. Behaviorally, light activation of the ACC-ACC connection facilitated behavioral withdrawal responses to mechanical stimuli and noxious heat. In an animal model of neuropathic pain, light inhibitory of ACC-ACC connection reduces both primary and secondary hyperalgesia. Our findings provide strong direct evidence for the excitatory or facilitatory contribution of ACC-ACC connection to pain perception, and this mechanism may provide therapeutic targets for future treatment of chronic pain and related emotional disorders.

[1] Center for Neuron and Disease, Frontier Institute of Science and Technology, Xi'an Jiaotong University, Xi'an, Shaanxi 710049, China. [2] Department of Physiology, Faculty of Medicine, University of Toronto, Medical Science Building, 1 King's College Circle, Toronto, Ontario M5S 1A8, Canada. [3] Institute of Brain Research, Qingdao International Academician Park, Qingdao, Shandong 266000, China. [4] CAS Key Laboratory of Brain Connectome and Manipulation, Interdisciplinary Center for Brain Information, The Brain Cognition and Brain Disease Institute, Shenzhen Institute of Advanced Technology, Chinese Academy of Sciences, Shenzhen-Hong Kong Institute of Brain Science Shenzhen Fundamental Research Institutions, Shenzhen, Guangdong 518055, China. [5] Department of Anesthesiology, Beijing Shijitan Hospital, Capital Medical University, 10th Tieyi Road, Haidian District, Beijing 100038, China. [6] Department of Neurology, First Affiliated Hospital of Guangzhou Medical University, Guangzhou, Guangdong 510130, China. [7] These authors contributed equally: Xu-Hui Li, Wantong Shi. ✉email: min.zhuo@utoronto.ca

Human and animal studies consistently demonstrate that excitation or potentiation in several cortical areas including the anterior cingulate cortex (ACC) and insular cortex play important roles in chronic pain-related emotional anxiety, behavioral sensitization, and pain unpleasantness[1–5]. The ACC is activated by noxious sensory stimuli[6,7] and anxiety disorders[1,8,9], and inhibiting the activities of the ACC produces analgesic effects and anxiolytic effects in humans and animals[6,10,11]. The ACC neurons receive sensory projections from the thalamus and somatosensory cortex[2,5,12,13], and the ACC contributes to the modulation of pain, and empathy of pain through its projections to other subcortical areas and spinal cord[14,15]. For its projection to the spinal cord dorsal horn, it has been reported that activation of ACC facilitates spinal nociceptive transmission, and this top-down facilitation contributes to behavioral hyperalgesia after peripheral nerve injury[16]. A recent study shows that ACC neurons send their direct projection to the nucleus accumbens (NAc) and this excitatory projection contributes to the empathy of pain[14]. Therefore, it is likely that ACC may contribute to distinct physiological and pathological functions through its distinct projections to different cortical and subcortical areas.

The corpus callosum is the major neural pathway that connects the two cerebral hemispheres in mammals, which consists of a network of cortical-cortical axonal fibers including the ACC-ACC connections fibers[17–19]. Previous clinical research shows that cortical-cortical connections play important in high brain functions, such as autism spectrum disorders, cognitive deficits, and epilepsy[17,20,21]. Previous in vivo animal studies found that ACC exerts excitatory influences on other side ACC neurons, and such connection may contribute to contralateral hindpaw pain[19,22]. However, there are few basic studies to characterize the physiological properties of such connections between ACCs, including the neurotransmitter and postsynaptic glutamatergic receptors that mediate the connection.

Our previous studies found that a single-digit amputation caused bilateral expression of immediate early genes in the ACC, bilateral loss of long-term depression, and bilateral synaptic potentiation[22,23]. These data suggest that ACC neurons receive inputs from both sides of the body, and the dense connecting fibers between the bilateral ACC may provide an additional mechanism for bilateral changes in the ACC. In the present studies, we like to examine the synaptic mechanisms between ACC and ACC pyramidal cells; and characterize synaptic connections to both the superficial layer (II and III) and deeper layer (V/VI) cells in the ACC. Furthermore, using optogenetic techniques, we were able to selectively activate the projecting connection between ACCs, and investigate their effects on behavioral nociceptive, emotional, and social interactions.

## Results

### Mapping the projection pathway from the ACC to the contralateral ACC

To verify the projection pathway from the ACC to the contralateral ACC, we used the rabies virus (RV)-based retrograde monosynaptic tracing strategy and the Volumetric Imaging with Synchronized on-the-fly-scan and Readout (VISoR) imaging technique. Here, we injected AAV-hSyn-EGFP-TVA-RVG as a helper virus into the ACC in the right hemisphere of the brain. The rabies virus (RV)-EnvA-ΔG-DsRed virus was injected into the same site of ACC after the AAV helper expression for 21 days. After 7 days of RV injection, the mice were perfused and imaged (Fig. 1a, b). Using VISoR imaging and 3D reconstruction, we obtained images of the whole brain and different sections of the mouse unilateral ACC projection inputs and outputs (Fig. 1c–f). We labeled EGFP-infected cells, DsRed-infected input cells, and starter cells (the initial RV-infected cells) infected with both EGFP and DsRed in the ipsilateral ACC of the virus injection. In the ipsilateral side, ACC neurons receive various brain regions ascending input, including the prelimbic cortex, retrosplenial cortex (RSC), motor cortex, thalamus, basolateral amygdala, hippocampus, and so on. On the contralateral side, the DsRed-infected input neurons were mainly distributed in the ACC and partly distributed in the prelimbic cortex, RSC, motor cortex, and BLA. In the ACC, the contralateral projection neurons were both in superficial (II/III) and deep (V/VI) layers of the ACC (Fig. 1f). In addition, the EGFP-labeled anterograde fibers were also observed in the contralateral ACC (Fig. 1e). These results provide evidence for a direct projection of the ACC to the contralateral ACC.

To intuitively observe the fibers of ACC-ACC projection, diluted AAV2/9-CMV-Cre was injected into the left ACC, and AAV2/R-EF1α-DIO-EYFP-EYFP was injected into the right ACC (Fig. 1g). Sparse labeling of contralateral projection neurons was achieved by combining VISoR and 3D reconstruction. We can observe labeled neurons in the left ACC and projective fibers from ipsilateral (left) ACC to contralateral (right) ACC crossing the corpus callosum (Fig. 1h, i).

To further verify the projection of the ACC-ACC is bilateral, we injected the anterograde viruses of AAV-CaMKIIα-EGFP and AAV-CaMKIIα-mCherry to the left and right ACC in mice, respectively (Fig. 1j). We found that both the EGFP- and mCherry- labeled anterograde fibers were observed in contralateral ACC (Fig. 1k, l), suggesting that the projection between two sides of ACC is reciprocal.

### Light stimulating evoked excitatory synaptic transmission in ACC-ACC projection

Next, we tested whether there is a direct synaptic transmission between the two sides of ACC. The optogenetic approach and whole-cell patch-clamp recording were combined to test the light-stimulating synaptic transmission between the bilateral ACC (Fig. 2a). In the injection side of ACC neurons, light stimulation can induce action potentials in the current clamp configuration with single, 5 shocks at 5 Hz, 10 shocks at 10 Hz, and 20 shocks at 20 Hz (Fig. 2b). We then patched contralateral pyramidal neurons and light stimulated the virus-infected fibers. The excitatory postsynaptic currents (EPSCs) were recorded in contralateral ACC neurons after repetitive light stimulation in the virus-infected fibers with 5 shocks at 5 Hz, 10 shocks at 10 Hz, and 20 shocks at 20 Hz (Fig. 2c). These synaptic responses followed the repetitive light stimulation without failure, suggesting that there are direct projections between the bilateral ACC. In addition, the amplitudes of light-evoked EPSCs increased in a stimulus intensity-dependent manner in the input (stimulation intensity)-output (evoked EPSCs amplitude) (I-O curves) test (Fig. 2d).

To test whether the excitatory synaptic transmission is mediated by glutamate receptors, the AMPA/kainate (KA) receptor antagonist 6-cyano-7-nitroquinoxaline-2, 3-dione (CNQX, 20 μM) and NMDA receptor antagonist D-2-amino-5-phosphonopentanoic acid (AP5, 50 μM) were bath applied. The EPSCs were rapidly and largely blocked by following application of AP5 and CNQX (Baseline: $100 \pm 8.1\%$; CNQX: $23.2 \pm 3.1\%$ of baseline; CNQX + AP5: $10.3 \pm 2.5\%$ of baseline; $n = 6$ neurons/4 mice; Fig. 2e). These results suggest that there is directly glutamatergic excitatory synaptic transmission in the ACC-ACC pathway.

### AMPA and KA receptors contribute to synaptic transmission in ACC-ACC projection

To further explore characteristics of synaptic transmission in the ACC-ACC projection, contralateral

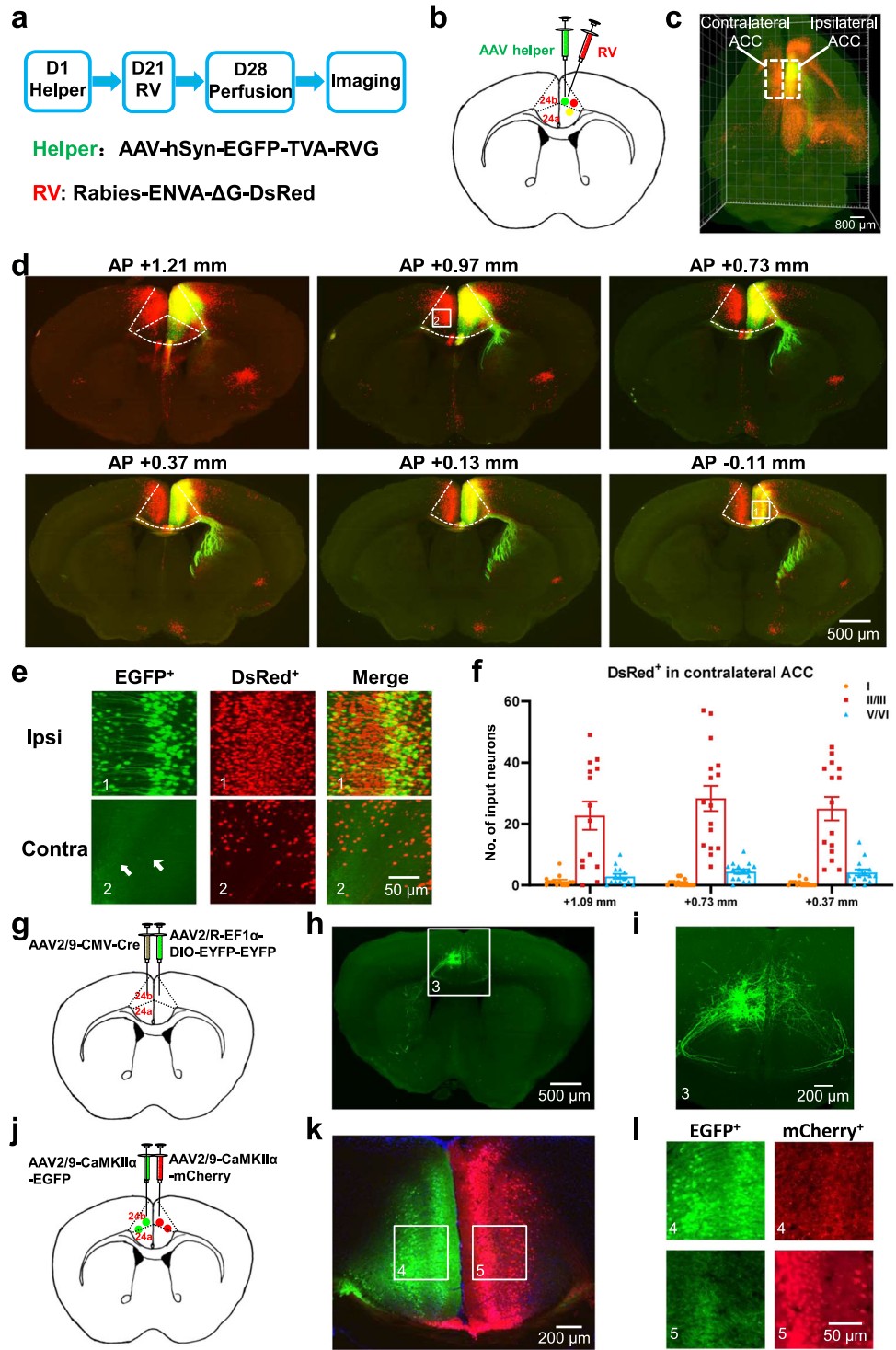

electrical stimulations were applied with a bipolar tungsten electrode placed in layer V of the ACC (Fig. 3a, b). To investigate whether the projections fibers between right and left ACC via the corpus callosum, three areas were stimulated as shown in Fig. 3a. There was a longer latency time and smaller EPSCs amplitude from stimulation in layer V of contralateral ACC compared with stimulation in the bilateral corpus callosum (Fig. 3c). To examine synaptic responses, we recorded the input (stimulation intensity)-output (EPSC amplitude) (I-O curves) relationship of EPSCs by stimulation in the layers V/VI or layers II/III of contralateral ACC. We found the amplitudes of these EPSCs increased with greater intensity of stimulation (Fig. 3d) ($n = 8$ neurons/4 mice).

Next, we tested monosynaptic responses by delivering repetitive stimulation of 5 shocks at 5 Hz, 10 shocks at 10 Hz, 20 shocks at 20 Hz, and the contralateral neurons that receive pure monosynaptic synaptic input (Fig. 3e). To test whether the glutamatergic excitatory synaptic transmission is mediated by KA receptor, we bath applied the potent AMPA receptor antagonist GYKI53655 (100 μM) and an AMPA/KA receptor antagonist, CNQX in the present of NMDA receptor antagonist AP5 in the bath solution. As shown in Fig. 3f, EPSCs were rapidly and rigorously blocked by GYKI 53655 in the neuron. Small residual EPSCs persisted in the presence of GYKI 53655 for 10 min after perfusion. Perfusion of CNQX entirely blocked the residual GYKI

**Fig. 1 Mapping of ACC-ACC projection pathway. a** The procedure of the retrograde trans-monosynaptic rabies virus (RV) tracing strategy. **b** A schematic of the AAV helper and RV separately micro-injected into the right ACC. **c** Fluorescence image of the whole brain after 3D reconstruction. The horizontal view for one 3D-reconstructed whole brain (red: DsRed$^+$; green: EGFP$^+$). Inside the dotted white box is the ACC. Scale bar, 800 μm. **d** Fluorescence images of the brain slice containing ACC (dotted white box) after injection of AAV helper and RV into the right ACC. Box 1 (ipsilateral ACC) and 2 (contralateral ACC) were augmented in (**e**). Scale bar, 500 μm. **e** Representative images of EGFP$^+$ and DsRed$^+$ neurons in ipsilateral (upper) and contralateral ACC (bottom) with the unilateral ACC injection of AAV helper and RV. The white arrows point to the projection fibers from the contralateral ACC. Scale bar, 50 μm. **f** Cell numbers analysis of DsRed$^+$ neurons at different layers of the contralateral ACC slices of the 30 μm thickness. **g** Schematic of left ACC injection of AAV2/9-CMV-Cre and right ACC injection of AAV2/R-EF1α-DIO-EYFP-EYFP in mice. **h** Fluorescence images of the brain slices from ACC-ACC sparsely labeled. The white box was augmented in Fig. 1i. Scale bar, 500 μm. **i** Representative image of labeled EYFP$^+$ neurons in the ipsilateral ACC and EYFP$^+$ projection fibers in the contralateral ACC. Scale bar, 200 μm. **j** A schematic of the left ACC injection of AAV-CaMKIIα-EGFP and right ACC injection of AAV-CaMKIIα-mCherry in mice. **k** Fluorescence images of the brain slices containing ACC after injection of AAV-CaMKIIα-EGFP and AAV-CaMKIIα-mCherry into ACC. White boxes were augmented in (**l**). Scale bar, 200 μm. **l** Representative images of labeled EGFP$^+$ neurons and mCherry$^+$ fibers in the left ACC (upper, box 4) and labeled mCherry$^+$ neurons and EGFP$^+$ fibers in the right ACC (bottom, box 5). Scale bar, 50 μm. Error bars represent SEM. ACC anterior cingulate cortex, Ispi ipsilateral, Contra contralateral.

53655-resistant current, suggesting that the current was mediated by KA receptors. However, we also found the evoked EPSCs in some neurons (4 neurons in total 10 neurons) by contralateral stimulation can be completely blocked by GYKI 53655. These results suggest that there is pure AMPA receptor-mediated synaptic transmission in the ACC-ACC synapse.

**Stimulation unilateral ACC evoked postsynaptic Ca$^{2+}$ influx in contralateral ACC neuron.** Calcium signaling is critical for synaptic transmission and plasticity in the ACC[1,2,24,25]. By using two-photon Ca$^{2+}$ imaging observation, our recent studies have characterized the properties of postsynaptic calcium signals in the pyramidal neurons of ACC in mice[26]. In the present study, by combining whole-cell patch recording and two-photon Ca$^{2+}$ imaging observation, we recorded the global Ca$^{2+}$ signals in the pyramidal neurons of layer V of the ACC in the slice by stimulating the contralateral ACC. After 30 min diffusion of Alexa594 K$^+$ salt and Cal-520 K$^+$ salt, the neuronal morphology was well labeled (Fig. 4a–c). Subthreshold stimulation in the ACC or corpus callosum can distinctly induce EPSCs in contralateral ACC neurons with different frequency stimulation (10, 50, and 100 Hz). However, it failed to record significant global Ca$^{2+}$ influx associated with the occurrence of five traces of EPSCs in contralateral neurons (Fig. 4d). The reason may be due to the contralateral subthreshold stimulation insufficient to increase the soma Ca$^{2+}$ influx in the ACC neuron. Next, a strong suprathreshold stimulation was applied to induce action potentials (APs) in the contralateral ACC neuron. We found that global calcium transients in contralateral neurons were observed when five AP spikes occurred (Fig. 4e). The ΔF/F values of Ca$^{2+}$ signals were increased with different stimulus frequencies (10, 50, and 100 Hz). These results suggest that activation of unilateral ACC evokes postsynaptic Ca$^{2+}$ influx in contralateral ACC neurons.

**Activated ACC-ACC pathway reduces nociceptive thresholds on the physiological condition.** Next, we want to employ the behavioral functions of the activity of ACC-ACC pathways. The optogenetic approach was used to test the light activation of the ACC-ACC pathway. We injected the AAV-hSyn-ChR2 (H134R)-EYFP virus into unilateral ACC and implanted optical fiber into the contralateral ACC and tested whether activation of the ACC-ACC projecting neurons would affect the nociceptive behaviors (Fig. 5a). We found that the mechanical withdrawal thresholds of both the left hind paw and right hind paw were significantly decreased when the left ACC to right ACC pathway was stimulated by blue light (465 nm), in comparison with the thresholds in light-off condition (Fig. 5b). In hot plate test, light activation of the ACC-ACC pathway significantly reduced the latency of response (Fig. 5c). In the tail-flick test, we found that during the

blue light (465 nm) "on" session, the tail flick withdrawal thresholds were significantly decreased after light excitation 10 min and will recover to baseline lever after 30 min (Fig. 5d). In addition, light activation of the ACC-ACC pathway has no significant effect on motor performance (Fig. 5e).

The ACC is well known to be involved in pain-related anxiety. We then performed an elevated plus maze (EPM) and open field test to observe anxiety-like behavior after light excitation ACC-ACC pathway (Fig. 5f). In the EPM test, during the blue light (465 nm) "on" session, the time in the open arm was significantly decreased compared with the light "off" session. The travel distance and the number of entries to the open arm were not changed after light excitation ACC-ACC projection (Fig. 5g, i). In the open field test, the light excitation ACC-ACC pathway has not significantly attenuated anxiety-like behaviors (Fig. 5h, j). In addition, to verify whether activation of the ACC-ACC pathway contributes to aversive behavior, a conditioned place avoidance (CPA) test was performed on mice after light-activating the ACC-ACC pathway (Fig. 5k). We found that light-activation of the ACC-ACC pathway during conditioning did not affect CPA score in mice (EYFP: 19.6 ± 18.0 s, $n = 5$ mice; ChR2: 5.2 ± 19.1 s, $n = 8$ mice) (Fig. 5l). Interestingly, light activation of the ACC-ACC pathway enhanced a conditioning CPA score at 7 days after nerve injury in male mice (EYFP: −14.9 ± 7.8 s, $n = 5$ mice; ChR2: −76.8 ± 12.2 s, $n = 7$ mice; $P < 0.01$). These findings indicate that an activated ACC-ACC pathway can reduce nociceptive thresholds and may partly contribute to affective behaviors.

To investigate possible gender differences, we also tested whether activation of the ACC-ACC pathway would affect nociceptive and affective behaviors in female mice. We found, consistent with the results in male mice, that when blue light (465 nm) stimulated the ACC-ACC pathway, mechanical withdrawal thresholds for the left and right hind paw were significantly reduced (Fig. 6a). In the hot plate test, activation of the ACC-ACC pathway by light significantly reduced the response latency (Fig. 6b). However, light activation of the ACC-ACC pathway during conditioning did not affect CPA score in female mice (EYFP: −60.4 ± 14.1 s, $n = 6$ mice; ChR2: −26.8 ± 8.9 s, $n = 9$ mice) (Fig. 6c). In the EPM test, during the blue light (465 nm) "on" session, the time in the open arms and the number of entries to the open arms were significantly decreased compared with the light "off" session. The travel distance has no significant difference after light excitation ACC-ACC projection (Fig. 6d, f). In the open field test, during the blue light (465 nm) "on" session, the travel distance, the time in the central zone, and the number of entries to the central zone have no significant difference after light excitation ACC-ACC projection (Fig. 6e, g). In summary, these findings indicate that the activated ACC-ACC pathway facilitates nociceptive responses in both adult male and female mice.

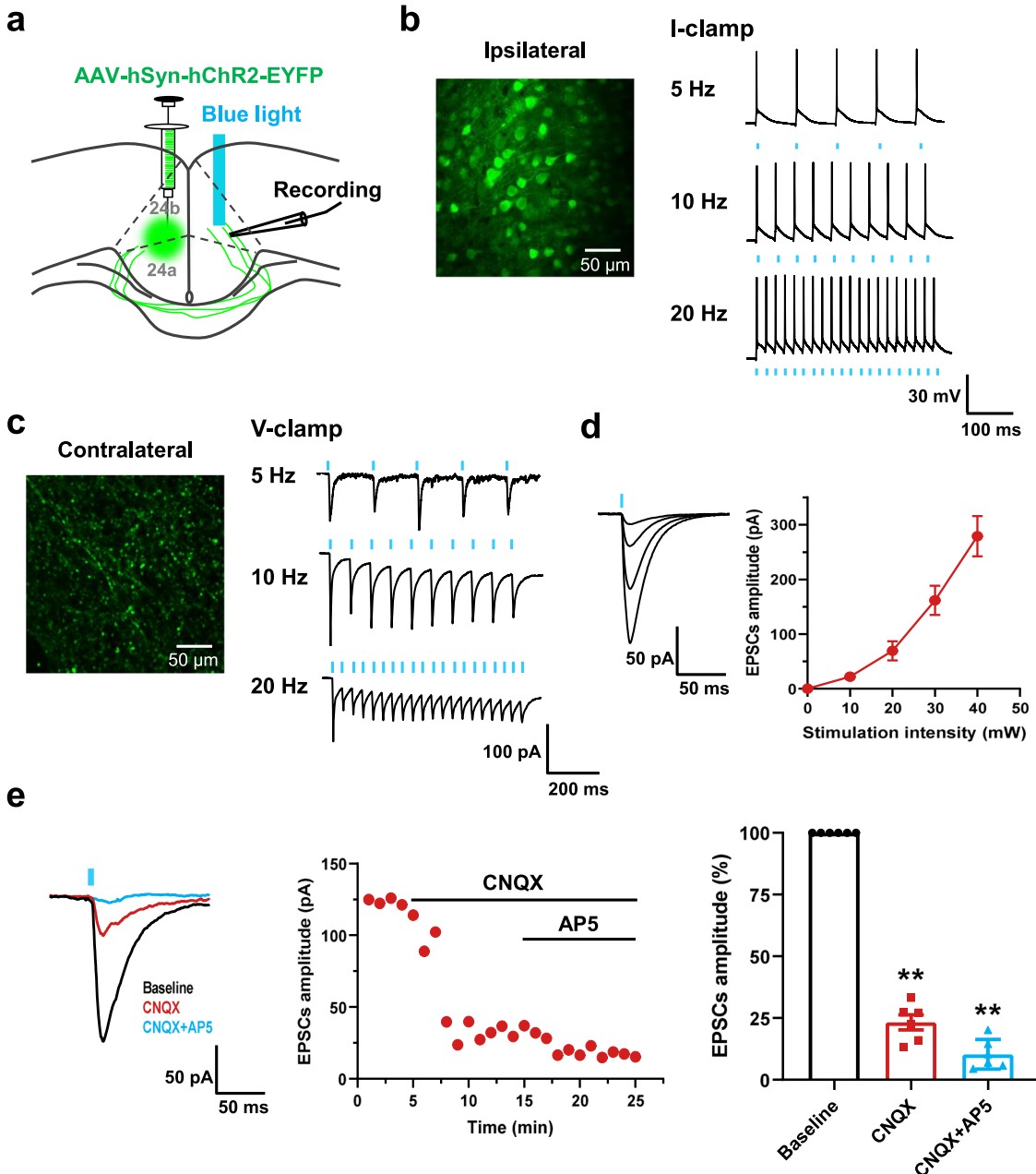

**Fig. 2 Optogenetic stimulation of ACC-ACC projection evoked glutamatergic excitatory synaptic transmission. a** A schematic diagram showing the placement of injecting virus in the one side ACC and the light stimulating and neurons recorded in the contralateral ACC. **b** Photograph of labeled neuron in the ipsilateral ACC and action potentials (APs) evoked by light stimulation at the 5, 10, and 20 Hz in the ipsilateral ACC neuron, scale bar, 50 μm. **c** A photograph showed the projection fibers in the contralateral ACC and light stimulation these fibers can evoked EPSCs on the contralateral ACC neurons at 5, 10, and 20 Hz. scale bar, 50 μm. **d** Input-output curves of blue light on projection fibers evoked EPSCs in contralateral ACC neurons (*n* = 8 neurons). **e** Applied CNQX and AP5 together can block light-evoked EPSCs in the ACC neuron (*n* = 8 neurons). Error bars represent SEM, *indicates a significant difference (**\**p* < 0.01). ACC anterior cingulate cortex, EPSCs excitatory postsynaptic currents.

**Activated the unilateral ACC reduces nociceptive thresholds of the contralateral hind paw in mice model of spared nerve injury**. We used optogenetic approaches to determine the effect of the ACC on behavioral pain responses in both hind paws of mice. After 3 weeks of AAV-hSyn-ChR2-EYFP expression and fiber embedding in mice with bilateral ACC, we tested whether activation of unilateral ACC affected nociceptive response in mice (Fig. 7a–c). We found that the mechanical withdrawal thresholds of the left hind paw were significantly decreased when the right ACC was stimulated by blue light (465 nm), and the mechanical withdrawal thresholds of the right hind paw were significantly

decreased when the left ACC was stimulated by blue light in comparison with the thresholds in light-off condition (Fig. 7b). In the hot plate test, light activation of the left or right ACC significantly reduced the latency of response (Fig. 7c). Next, we tested whether inhibition of unilateral ACC affected nociceptive behaviors (Fig. 7d–f). For mice with AAV-hSyn-eNpHR-mCherry injected into the bilateral ACC, yellow light (593 nm) suppression of the right ACC increased the mechanical withdrawal threshold of the left hind paw, and inhibition of the left ACC increased the mechanical withdrawal threshold of the right hind paw (Fig. 7e). Yellow-light inhibition of the left or right

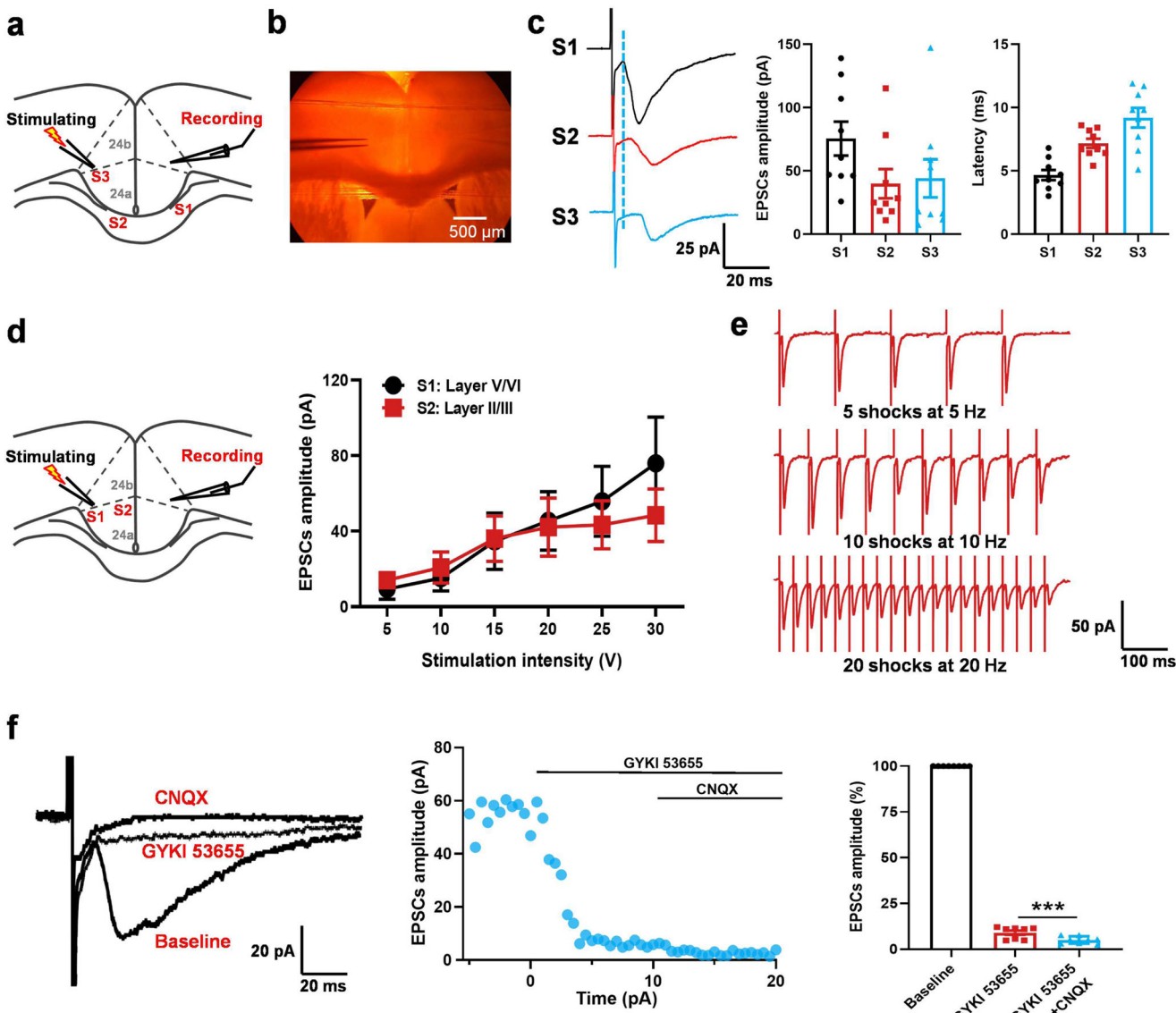

**Fig. 3 AMPA/KA receptors mediated glutamatergic excitatory synaptic transmission in ACC-ACC synapses. a, b** Representative recording diagram and photograph showing the placement of stimulating electrode in one side of ACC or corpus callosum and recording electrode in the contralateral ACC. S1-3 represents different stimulus sites. **c** Representative traces (left), amplitude (middle), and latency (right) of EPSCs in the contralateral ACC at different stimulus sites of S1-3 ($n = 10$ neurons/5 mice). **d** Left: representative recording diagram showing the placement of stimulating electrode on the one side of ACC and recording electrode in the contralateral ACC. S1 and S2 represent stimulation sites in layers V/VI and II/III of ACC, respectively. Right: the input-output relationship of the contralateral ACC EPSCs when the S1 or S2 sites of one side ACC are stimulated ($n = 8$ neurons/4 mice). **e** The monosynaptic responses by delivering 5 shocks at 5 Hz, 10 shocks at 10 Hz, and 20 shocks at 20 Hz. **f** In the presence of picrotoxin (100 μM) and AP5 (50 μM), KA receptor-mediated EPSCs could be observed after the application of GYKI 53655 (100 μM) and then blocked by CNQX (20 μM). Sample traces (left), sample time course points (middle), and statistical results (right) showed the EPSCs in the presence of GYKI 53655 and CNQX ($n = 6$ neurons/3 mice). Error bars represent SEM, *indicates a significant difference (***$p < 0.001$). ACC anterior cingulate cortex, EPSCs excitatory postsynaptic currents.

ACC increased the response latency of hot plate test in mice (Fig. 7f).

To determine the effect of unilateral ACC on the nociceptive response of both hind paws of mice after nerve injury, a spared nerve injury model was performed on the left hind limb of mice 3 weeks after virus expression and fiber embedding, followed by behavioral tests (Fig. 7g, h). After the nerve injury model, the mechanical withdrawal threshold of the ipsilateral (left) hind paw was significantly decreased (Fig. 7i). Blue light activation of left or right ACC did not affect the mechanical threshold of the ipsilateral hind paw after the nerve injury model (Fig. 7j). However, yellow light suppression of the right ACC decreased the

hyperalgesia of the ipsilateral hind paw 3, and 7 days after the nerve injury model (Fig. 7k). In addition, we also found that there was hyperalgesia in the contralateral (right) hind paw after 7 days nerve injury model (Fig. 7l). Activation of the left ACC by blue light reduced the mechanical threshold of the contralateral hind paw after 14 and 28 days of nerve injury (Fig. 7m). In addition, inhibition of the left ACC by yellow light decreased the hyperalgesia in the contralateral hind paw in nerve injury model mice (Fig. 7n). In summary, these results suggest that the activation of the unilateral ACC promotes a nociceptive response in the contralateral hind paws of nerve-injured mice, and inhibition can have an analgesic effect.

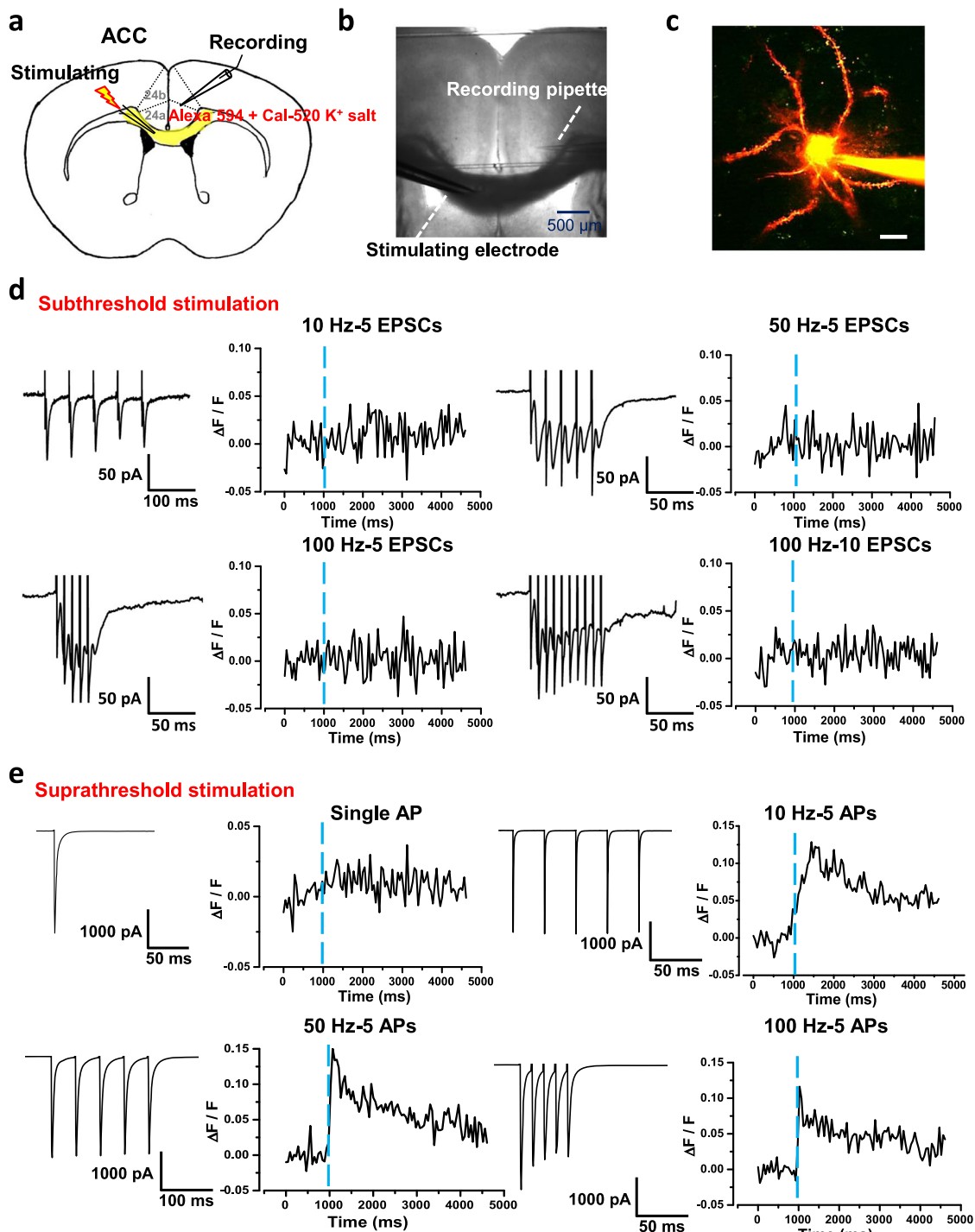

**Fig. 4 Stimulating one side of ACC can evoke postsynaptic Ca$^{2+}$ influx in contralateral ACC neurons. a, b** A representative recording diagram and a photograph showing the placement of the stimulating electrode in the corpus callosum and the recording electrode in the contralateral ACC. **c** A representative photomicrograph of a labeled pyramidal neuron in the contralateral ACC, scale bar, 20 μm. **d** The EPSCs and Ca$^{2+}$ influx in contralateral ACC neurons induced by subthreshold electric stimulation with 5 shocks at 10 Hz (upper left), 5 shocks at 50 Hz (upper right), 5 shocks at 100 Hz (bottom left), and 10 shocks at 100 Hz (bottom right). **e** The APs and Ca$^{2+}$ influx in contralateral ACC neurons induced by suprathreshold electric stimulation with single APs (upper left), 5 APs at 10 Hz (upper right), 5 APs at 50 Hz (bottom left), and 5 APs at 100 Hz (bottom right). ACC anterior cingulate cortex, EPSCs excitatory postsynaptic currents, APs action potentials.

**Inhibited the ACC-ACC pathway has analgesic effects in nerve-injured model mice.** The effect of inhibition of unilateral ACC showing analgesic effect has been discussed above. Next, we further explore the effect of inhibition of the ACC-ACC pathway on analgesic effects in nerve injury model mice. The AAV-hSyn-eNpHR-mCherry was injected into the right ACC and implanted optical fiber on the left ACC (Fig. 8a). We found that yellow light suppression of the ACC-ACC projection increased the mechanical withdrawal threshold of the right hind paw, and had no effect on the left hind paw in normal mice (Fig. 8b). However, the

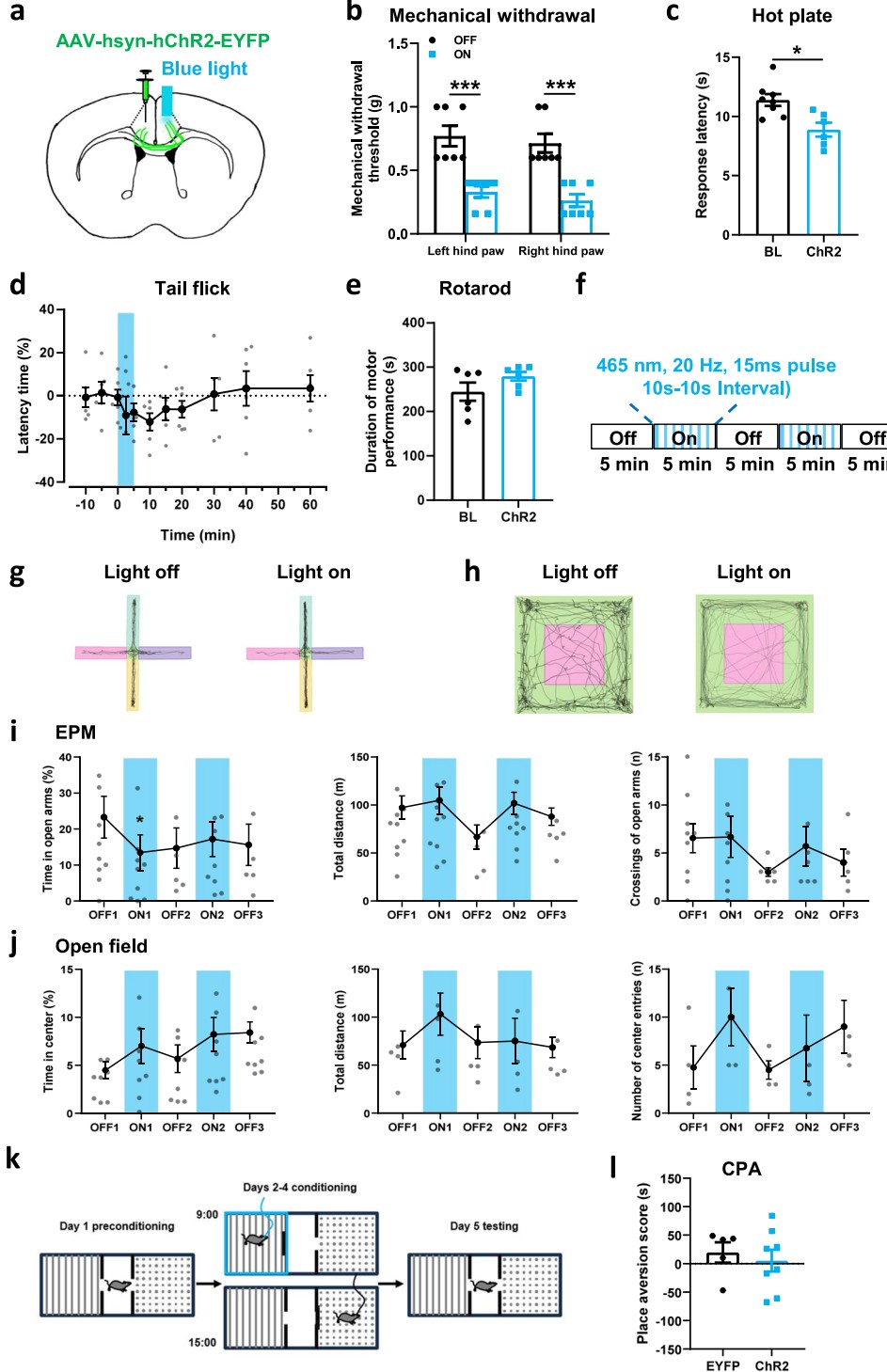

response latency in hot plate test had no significant change after yellow light suppression of the ACC-ACC pathway. (Fig. 8c). The nerve injury model was performed on the left hind limb of mice after virus expression for 3 weeks (Fig. 8d). Activation of ACC-ACC did not affect the mechanical withdrawal threshold of the ipsilateral (left) hind paw after nerve injury, and inhibition of ACC-ACC projection decreased the hyperalgesia (Fig. 8f, g). In addition, activation of ACC-ACC projection decreased the mechanical withdrawal threshold of the contralateral (right) hind paw after nerve injury, and inhibition of ACC-ACC projection had an analgesic effect after nerve injury (Fig. 8h–j). In summary, these results indicated that the activation of the ACC-ACC

projection promotes nociceptive behavior in nerve-injured mice, and inhibition can have an analgesic effect.

## Discussion
With the use of optogenetic experimental approaches, there is an increased investigation of ACC and its related brain regions in chronic pain and its related emotional disorders[27]. While synaptic potentiation in the ACC contributes to behavioral sensitization and anxiety in animal models of chronic pain[6,10,28], the nerve projections from ACC to other subcortical structures and spinal cord also exert descending facilitation of pain (spinal cord)[16] and behavioral fear (amygdala)[29–32] and empathy of pain

**Fig. 5 The effects of optogenetic stimulation ACC-ACC pathway on physiological behaviors. a** Schematic diagram showing the placement of virus injected in the one side ACC and the optic fiber implantation in the contralateral ACC. **b** Blue light activation of the ACC-ACC pathway significantly decreased left and right hind paw withdrawal threshold in mice ($n = 8$ mice, $p < 0.001$ for OFF versus ON of left hind paw; $p < 0.001$ for OFF versus ON of right hind paw). **c** Blue light activation of the ACC-ACC pathway significantly decreased the response latency of the hot plate test in mice ($n = 8$ mice, $p < 0.05$ for OFF versus ON). **d** Blue light activation of the ACC-ACC pathway significantly decreased the response latency of the tail flick test in mice ($n = 6$ mice). **e** Blue light activation of the ACC-ACC pathway did not affect motor function in mice ($n = 6$ mice). **f** A schematic diagram shows the timing and parameters of blue light given in the EPM and open field tests. **g** Representative traces for showing the movement of mice in the EPM test. The horizontal pink and purple boxes are open arms. The vertical yellow and green boxes are closed arms. **h** Representative traces for showing the movement of mice in the open field test. The pink box is the central area and the green is the peripheral area. **i** The anxiety behaviors (total distance, number of entries, and time in open arm) were observed in EPM with blue light-evoked activation of the ACC-ACC pathway ($n = 9$ mice). **j** After the activation of the ACC-ACC pathway by blue light, there was no significant difference in the total distance, the number, and time of entering the central region of mice in the open field ($n = 8$ mice). **k** The paradigm of conditioned place aversion (CPA) test. **l** After the activation of the ACC-ACC pathway by blue light, there was no significant difference in the place aversion score of the light-paired chamber compared with the EYFP group (EYFP, $n = 5$ mice; ChR2, $n = 8$ mice). Error bars represent SEM, *indicates significant difference (*$p < 0.05$, ***$p < 0.001$). ACC anterior cingulate cortex, EPM elevated plus maze.

(accumbens)[14]. In the present study, we characterized the excitatory glutamatergic connections between ACC and contralateral ACC pyramidal neurons. Activation of such excitatory connections by repetitive stimulation caused monosynaptic excitatory postsynaptic responses and produced increases in postsynaptic calcium signaling. Behaviorally, selective activation of the ACC-ACC excitatory connection triggered the facilitation of behavioral nociceptive responses to peripheral noxious stimuli. In addition, anxiety-like responses were also enhanced. Our preliminary studies found that such excitatory connections are also plastic, and may contribute to long-lasting changes in ACC-related cortical circuits in chronic pain and related disease conditions.

The communication between the two hemispheres is made possible by the corpus callosum (CC) and a diverse population of callosal projection neurons which play a critical role in higher-level associative connectivity (Table 1)[18,33]. Most callosal projection neurons are excitatory however there are reports of inhibitory neurons as well[34]. In the present study, we found that ACC pyramidal cells projected to neurons located in both Layer II/III and V at the contralateral ACC. By using electrophysiological, pharmacological, and optogenetic methods, we found that these callosal projections are mostly glutamatergic synapses. We did not detect oblivious inhibitory projections in our studies, although we cannot completely rule out this possibility. By recording from pyramidal cells located in different layers of ACC, we revealed that activation of callosal projections by electrical stimulation and/or optogenetic stimulation triggered typical fast synaptic responses in pyramidal cells in the contralateral ACC. Postsynaptic responses recorded from superficial layers are like those recorded from deep layers. We also found the projections to layer I and other possible interneurons, although these synapses are not the focus of our present study. Future experiments are needed to study the functions of these synapses. Neurons located in layers II/III and V are not isolated. It has been reported that many of them are connected by excitatory projections. Wu et al.[35] reported by the dual patch-clamp study found that ACC neurons within the superficial layers (II/III) can form direct excitatory connections with neurons in deep layer V, providing direct evidence that neurons located at both left and right side of ACC may form a chain of the excitatory network. A recent study found that some deep pyramidal sends descending projections to the spinal cord[16,36]. Our present study provides evidence that ACC may project to contralateral ACC pyramidal cells, and subsequently modulate spinal pain transmission through ACC-spinal cord projections.

Most excitatory synapses in the brain are glutamatergic. Recent studies consistently indicate that these excitatory synapses are heterogeneous[22,37]. There are at least three primary forms of synapses that are reported in adult synapses of the cortex. The first synapses are called silent synapses or pure NMDA receptor synapses. In these synapses, there is no functional AMPA or KA receptor available to detect the release of glutamate[37–39]. In a previous study of freely moving animals, Wu et al.[22] reported that ACC stimulation induced mixed fast and slow responses in the contralateral ACC. These findings indirectly suggest the possible existence of pure NMDA synapses. Furthermore, in the field recording of ACC responses, Chen et al.[40] also reported silent responses in the ACC that can be recruited after the induction of LTP. Second synapses are carried out by AMPA receptors. In the hippocampal CA1 region, synaptic responses are mediated by the AMPA receptor only. However, in the ACC, synapse responses induced by local stimulation are mediated by AMPA and kainate receptors, suggesting that these synapses are mixed with AMPA/KA receptors[41,42]. In the present study, we found that ACC-ACC synapses are mixed with AMPA and KA receptors, further supporting the fact that cortical synapses in the adult brain are heterogeneous. These unique properties may also allow adult synapses to undergo up and down-regulation.

Recent studies have found that ACC may contribute to different physiological functions through its selective projection to other cortical and subcortical structures. ACC to accumbens is associated with the empathy of pain[14]. ACC sends its direct projection to the spinal cord and facilitates the spinal transmission of nociceptive information[16]. In the present study, we found that ACC forms a connection with contralateral ACC by excitatory connections. Such unique pathways will allow ACC to influence pain transmission at the spinal cord level through the projection to the other side, and then descending projections (Fig. 9). In our results, we found that the mechanical withdrawal thresholds of both hindpaws were significantly decreased when the light-activating the projection fibers from left ACC. This may be due to the nature of bilateral projection modulatory systems to the spinal cord. We cannot rule out the possible contribution of activation of cell bodies by retrograde action potential from axon terminals with ChR2. Furthermore, the ACC-ACC connection reported here may also contribute to the bilateral modulation of pain from the cortex. We here propose that several possible positive loops exist in the CNS to enhance or facilitate pain-related information: short-distance positive feedback; long-distance positive feedback; unilateral positive feedback and bilateral positive feedback. Considering the circuit-like innervation between bilateral ACCs, such unique circuits provide a positive feedback loop for the processing of nociceptive information in the cortex. Thus, pain perception can be reinforced through such short-distance and long-distance projections.

Considering the important protective function of pain in the evaluation, such a link will allow animals and humans to have pain sensitivity even in case of injury to one side of the cortex. Future experiments are needed to know how the circuits are formed for ACC neurons to receive excitatory inputs from

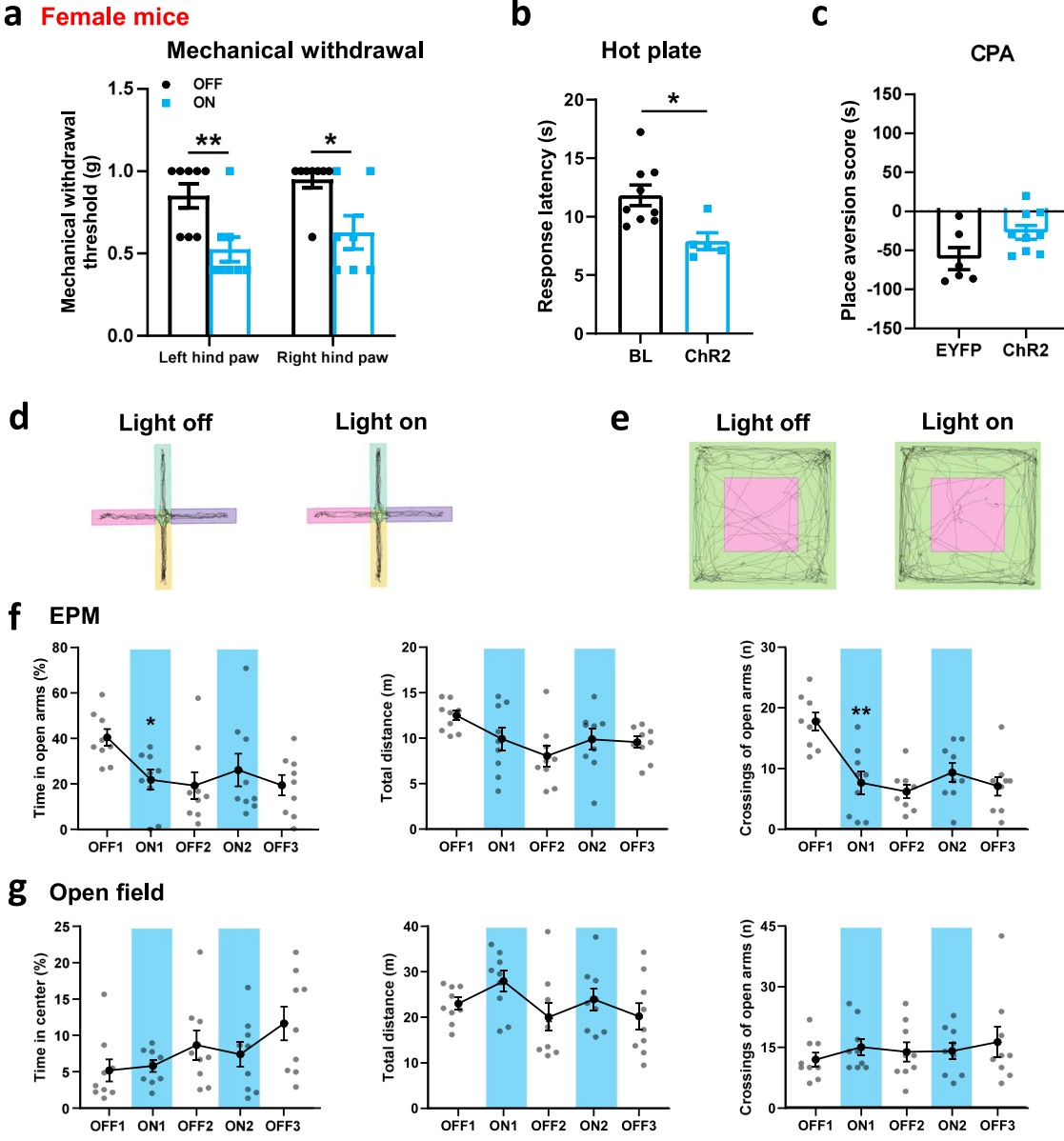

**Fig. 6 The effects of optogenetic stimulation ACC-ACC pathway on physiological behaviors in female mice. a** Blue light activation of the ACC-ACC pathway significantly decreased left and right hind paw withdrawal threshold in female mice ($n = 8$ mice, $p < 0.01$ for OFF versus ON of left hind paw; $p < 0.05$ for OFF versus ON of right hind paw). **b** Blue light activation of the ACC-ACC pathway significantly decreased the response latency of the hot plate test in female mice ($n = 9$ mice, $p < 0.05$ for OFF versus ON). **c** Blue light activation of the ACC-ACC pathway did not affect conditioned place aversion in female mice (EYFP, $n = 6$ mice; ChR2, $n = 9$ mice). **d** Representative traces for showing the movement of mice in the EPM test. The horizontal pink and purple boxes are open arms. The vertical yellow and green boxes are closed arms. **e** Representative traces for showing the movement of mice in the open field test. The pink box is the central area and the green is the peripheral area. **f** The anxiety behaviors (total distance, number of entries and time in open arm) were observed in EPM with blue light-evoked activation of the ACC-ACC pathway ($n = 9$ mice). **g** The anxiety behaviors (total distance, number of center entries, and time in center) were observed in an open field test with blue light-evoked activation of the ACC-ACC pathway ($n = 9$ mice). Error bars represent SEM, *indicates a significant difference (*$p < 0.05$, **$p < 0.01$). ACC anterior cingulate cortex, EPM elevated plus maze.

contralateral ACC as well as other areas, and then pass the information to ACC neurons that send top-down projection to the spinal-cord dorsal horn as well as other subcortical areas.

ACC is important in acute/physiological pain and chronic pain[2,5,24]. For acute pain, the lesion or inhibition in the ACC can inhibit behavioral responses to peripheral noxious stimuli[2,24,43,44]. Consistent, electrical stimulation of ACC facilitates spinal nociceptive responses and generates fearful memories[15,45]. Chemical activation of ACC can also induce aversive memory[46]. In this study, we found that activation of the ACC-ACC connection also produced the facilitation of behavioral

reflexive responses. These include both hind-paw withdrawals as well as a spinal nociceptive tail-flick reflex. While facilitation of the tail-flick reflex can be explained by top-down facilitation as previously reported[15,16], facilitation of the hind-paw withdrawal reflex may be due to supraspinal facilitation in the ACC as well as facilitation by the ACC-spinal projection system. These findings indicate that ACC may engage different projection pathways to facilitate pain perception in the brain or pain transmission at the spinal cord level. Our studies also indicate that ACC-ACC connectivity involves affective modulation, such as anxiety and aversive behaviors in pathological pain conditions. While

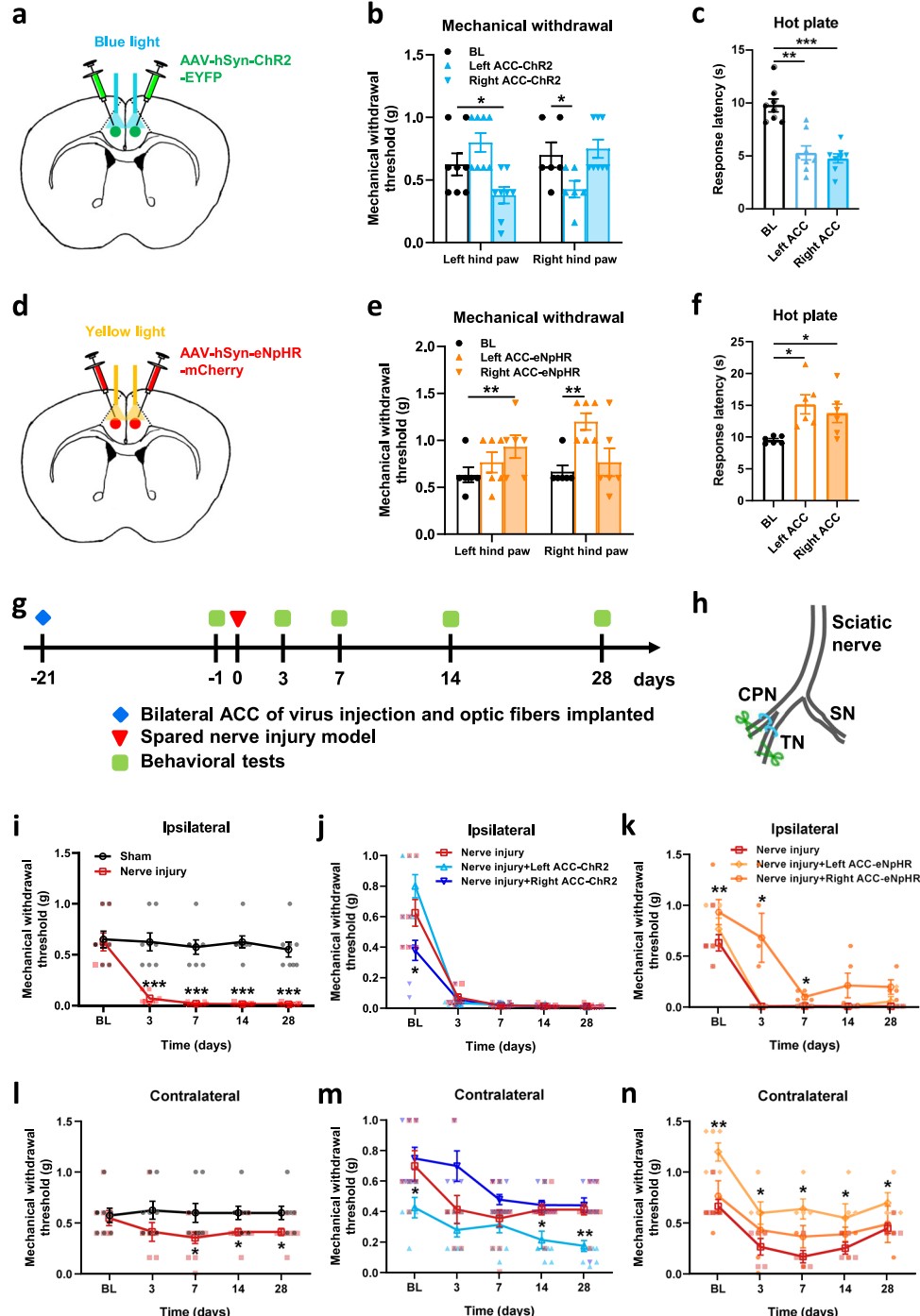

light-activating ACC-ACC connectivity does not significantly affect anxiety and aversive behaviors in the physiology condition, the same stimulation enhanced anxiety and aversive behaviors in animals with nerve injury. These findings indicate that the ACC-ACC connection provides fine modulation for somatosensory pain processing while contributing less to negative painful emotional responses. However, in case of nerve injury, this connection is also recruited and contributes to negative emotional and behavioral responses.

For chronic pain, it has been reported that cortical excitation contributes to behavioral allodynia and hyperalgesia[2,6,24]. In an animal model of nerve injury, ACC synaptic transmission was significantly enhanced through increased expression of postsynaptic AMPA receptors[2,40,47]. In addition, NMDA receptors

are also enhanced; and pharmacological inhibition of these receptors can reduce behavioral sensitization as well as pain-induced anxiety[1,10]. In this work, we added evidence that the ACC-ACC excitatory connection also contributes to neuropathic pain. Optogenetic inhibition of ACC-ACC produced a reduced significant reduction of behavioral sensitization. These results provide evidence that ACC-ACC glutamatergic projection contributes to chronic pain.

## Methods

**Animals.** Adult (8 to 12 weeks old) male or female C57BL/6 mice were purchased from the Experimental Animal Center of Xi'an Jiaotong University. All animals were randomly housed under an

**Fig. 7 Optogenetic activation of the ACC induced pain behavior in the contralateral hind paw in nerve-injured mice, whereas inhibition relieved pain.**
**a** Schematic of bilateral ACC injection of AAV-hSyn-ChR2-EYFP and implantation of optic fibers in mice. **b** Blue light activation of right ACC significantly decreased left hind paw withdrawal threshold, and activation of left ACC significantly decreased right hind paw withdrawal threshold in mice ($n = 8$ mice, $p < 0.05$ for BL versus right ACC-ChR2 of left hind paw, $p < 0.05$ for BL versus left ACC-ChR2 of right hind paw). **c** Blue light activation of left or right ACC significantly decreased response latency of hot plate test in mice ($n = 8$ mice, $p < 0.01$ for BL versus left ACC-ChR2, $p < 0.001$ for BL versus left ACC-ChR2). **d** Schematic of bilateral ACC injection of AAV-hSyn-eNpHR-mCherry and implantation of optic fibers in mice. **e** Yellow light inhibition of right ACC significantly increased the left hind paw withdrawal threshold, and inhibition of left ACC significantly increased the right hind paw withdrawal threshold in mice ($n = 6$ mice, $p < 0.01$ for BL versus right ACC-eNpHR of left hind paw, $p < 0.01$ for BL versus left ACC-eNpHR of right hind paw). **f** Yellow light inhibition of left or right ACC significantly increased response latency of hot plate test in mice ($n = 6$ mice, $p < 0.05$ for BL versus left ACC-eNpHR, $p < 0.05$ for BL versus left ACC-eNpHR). **g** Experimental timeline of the nerve injury and behavior tests in mice. **h** Schematic diagram of the pain model of spared nerve injury in mice. **i** Compared with the sham group, the mechanical threshold of the ipsilateral hind paw with injury was significantly reduced in the nerve injury group (Sham, $n = 8$ mice, Nerve injury, $n = 11$ mice, for Sham versus Nerve injury, $p < 0.001$ for 3 d, 7 d, 14 d, and 28 d). **j** Blue light evoked-activation of ipsilateral or contralateral ACC after nerve injury did not affect the withdrawal threshold of the ipsilateral hind paw ($n = 11$ mice). **k** Yellow light inhibition of right ACC significantly increased the withdrawal threshold of the ipsilateral hind paw 3, and 7 days after nerve-injured in mice ($n = 5$ mice, for Nerve injury versus Nerve injury + right ACC-eNpHR, $p < 0.05$ for 3d and 7 d). **l** Compared with the sham group, the mechanical threshold of the contralateral hind paw with injury was significantly reduced in the nerve injury group (Sham, $n = 8$ mice; Nerve injury, $n = 11$ mice, for Sham versus Nerve injury, $p < 0.05$ for 7 d, 14 d, and 28 d). **m** Blue light activation of ipsilateral ACC significantly decreased the contralateral hind paw withdrawal threshold 14, and 28 days after nerve-injured in mice ($n = 11$ mice, for Nerve injury versus Nerve injury + left ACC-ChR2, $p < 0.05$ for 14 d, $p < 0.01$ for 28 d). **n** Yellow light inhibition of ipsilateral ACC significantly increased the withdrawal threshold of the contralateral hind paw 3, 7, 14, and 28 days after nerve-injured in mice ($n = 5$ mice, for Nerve injury versus Nerve injury + left ACC-eNpHR, $p < 0.05$ for 3 d, 7 d, 14 d, and 28 d). Error bars represent SEM, *indicates significant difference (*$p < 0.05$, **$p < 0.01$, ***$p < 0.001$). ACC anterior cingulate cortex, BL baseline.

---

artificial 12-hour light/dark cycle (7 a.m. to 7 p.m. light) in a holding room kept at a temperature of $23 \pm 1\,°C$ with food and water provided ad libitum. All experiment protocols have been approved by the Ethics Committee of Xi'an Jiaotong University.

**Drug application.** All the chemicals and drugs were purchased from Tocris Cookson (Bristol, UK). AP5 (Cat. No.0106) is a selective competitive NMDA receptor antagonist, it was prepared in distilled water. Selective non-NMDA ionotropic glutamate receptor antagonist CNQX (Cat. No.0190) was dissolved in dimethyl sulfoxide (DMSO). AMPA receptor antagonist GYKI53655 (Cat. No.2555) was dissolved in distilled water. $GABA_A$ receptor antagonist Picrotoxin (Cat. No.1128) was dissolved in ethanol as a stock solution. All these stock solutions were diluted to the final desired concentration in the artificial cerebrospinal fluid (ACSF) before immediate use.

**Virus injection.** For trans-monosynaptic retrograde tracing experiments, 200 nL of AAV2/9-hSyn-EGFP-2a-TVA-2a-RVG-WPREs-pA ($2.0 \times 10^{12}$ genomics copies per mL, Brainvta, Wuhan, China) was injected into the right ACC, and 200 nL of RV-EnvA-ΔG-DsRed ($2.0 \times 10^8$ genomic copies per mL, Brainvta, Wuhan, China) was injected into the same site three weeks after AAV viral expression. For anterograde tracing experiments, 150 nL of AAV2/9-CaMKIIα-EGFP-WPRE-hGH-pA ($2.0 \times 10^{12}$ genomics copies per mL, Brainvta, Wuhan, China) and AAV2/9-CaMKIIα-mCherry-WPRE-hGH-pA ($2.0 \times 10^{12}$ genomics copies per mL, Brainvta, Wuhan, China) were separately injected into the left and right ACC. For ACC-ACC single neuron was labeled, 50 nL of AAV2/9-CMV-Cre ($2.0 \times 10^{12}$ genomics copies per mL, Brainvta, Wuhan, China, dilute 1000 times) and AAV2/R-EF1α-DIO-EYFP-EYFP ($2.0 \times 10^{12}$ genomics copies per mL, Brainvta, Wuhan, China) were separately injected into the left and right ACC. For in vitro whole-cell patch-clamp recording, 200 nL of AAV2/9-hSyn-hChR2 (H134R)-EYFP-WPRE-hGH-pA ($1.2 \times 10^{12}$ genomics copies per mL, Brainvta, Wuhan, China) was injected into the unilateral ACC. For optogenetic stimulation of the ACC-ACC pathway, 200 nL of AAV2/9-hSyn-hChR2(H134R)-EYFP-WPRE-hGH-pA ($1.2 \times 10^{12}$ genomics copies per mL, Brainvta, Wuhan, China) or AAV2/9-hSyn-eNpHR-mCherry-WPRE-hGH-pA ($1.2 \times 10^{12}$ genomics copies per mL, Brainvta, Wuhan, China) was injected into the unilateral

ACC. For optogenetic stimulation of ACC, 200 nL of AAV2/9-hSyn-hChR2 (H134R)-EYFP- WPRE-hGH-pA ($1.2 \times 10^{12}$ genomics copies per mL, Brainvta, Wuhan, China) or AAV2/9-hSyn-eNpHR-mCherry-WPRE-hGHpA ($1.2 \times 10^{12}$ genomics copies per mL, Brainvta, Wuhan, China) was injected into the bilateral ACC.

Viral injection procedures were performed as previously described[28]. Briefly, mice were anesthetized with 2% isoflurane and placed in a stereotaxic device. Cut the skin to expose the skull and drill holes in the ACC location (0.9 mm anterior to bregma, 0.3 mm lateral to the midline, and 1.4 mm deep from the cerebral surface). The viruses were injected with equal speed by micro-syringe pump (23 nL/min, once every 10 seconds; Nanoject II #3-000-205/206, DRUMMOND) into the ACC location. After injection, the surgical wound was carefully sutured and disinfected. The animals were monitored until they woke up before returning to their home cages. Mice were allowed to survive for ~3 weeks for AAV virus expression and 7 days for RV virus expression.

**Brain separation and imaging.** For antegrade tracing and in vitro whole-cell patch-clamp recording experiments, the mice with the virus infection were deeply anesthetized with 2% iso-flurane and perfused intracardially with 0.01 M phosphate-buffered saline (PBS, pH 7.4) followed by 4% w/v paraf-ormaldehyde (PFA, pH 7.4). The whole brain was immediately removed and placed in 4% PFA solution for 4-h post-fixation. Then, the whole brain was stored in 0.1 M PB containing 30% (w/v) sucrose solution for 3-d dehydration. The brain was cut into 30 μm-thickness coronal brain slices using a freezing microtome (Leica CM1900). These sections were counterstained with DAPI (ABS9235, absin, Shanghai) and observed using a laser scanning confocal microscope (FV1000, Olympus, Japan) or a slide scanner (Slideview VS200, Olympus).

For trans-monosynaptic retrograde tracing experiments, we used a fast and high-resolution microscopy method of Volumetric Imaging with Synchronized on-the-fly-scan and Readout (VISoR) as previously described[48,49]. After perfusion, the brain was removed and incubated in 4% hydrogel monomer solution (HMS, 4% w/v acrylamide, 0.05% w/v bisacrylamide, 1× PBS, 4% w/v PFA, 0.25% VA-044 thermal initiator, and distilled water) at 4 °C for 2 days, to allow penetration of fixation solution. The

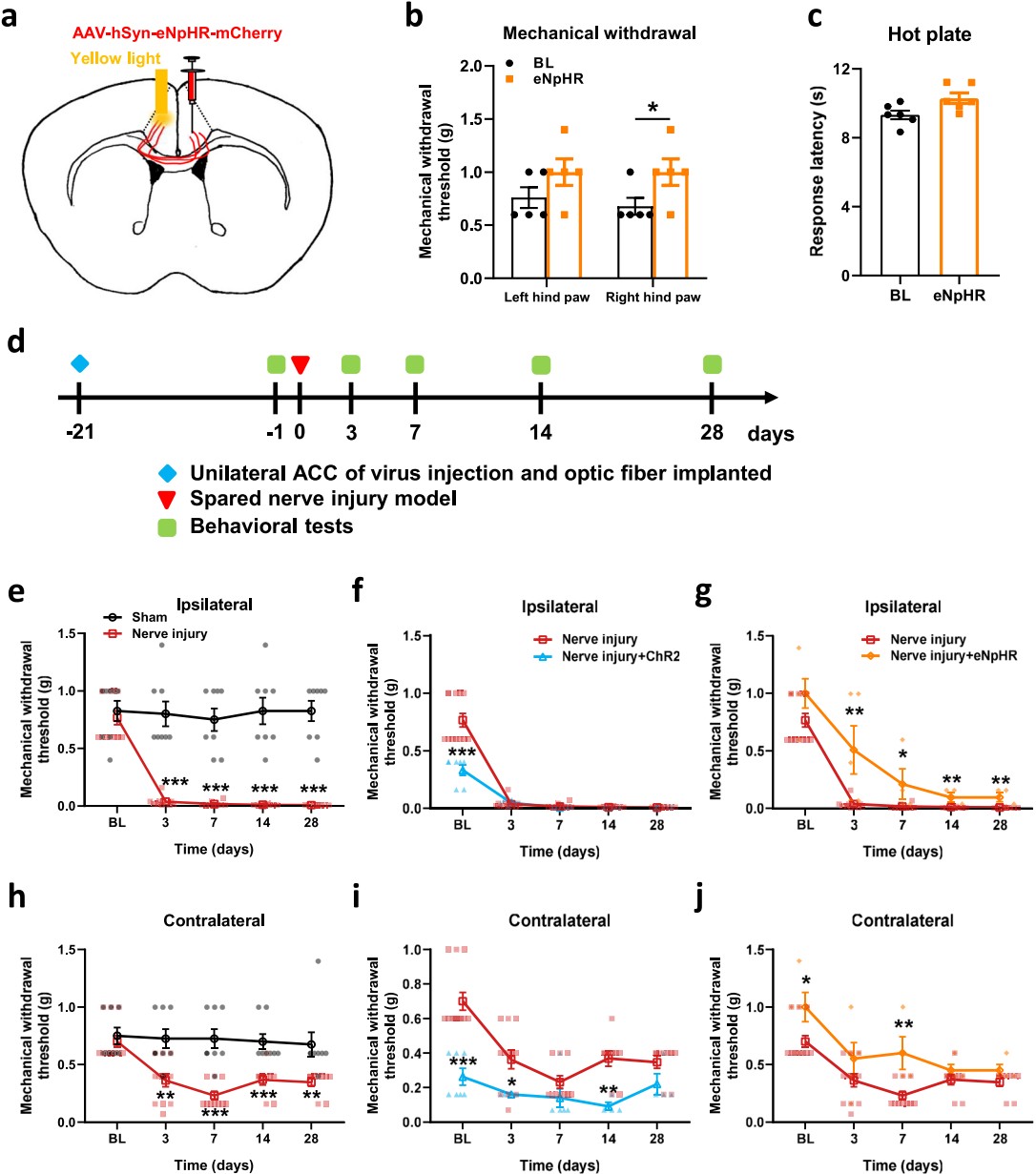

**Fig. 8 Optogenetic activation of the ACC-ACC induced pain behavior in the contralateral hind paw in nerve-injured mice, whereas inhibition relieved pain. a** A schematic diagram showing the placement of the virus injected in one side ACC and the optic fiber implantation in the contralateral ACC. **b** Yellow light inhibition of the ACC-ACC significantly increased the right hind paw withdrawal threshold ($n = 5$ mice, $p < 0.05$ for BL versus eNpHR of the right hind paw). **c** Yellow light inhibition of the ACC-ACC did not affect the response latency of the hot plate test in mice ($n = 5$ mice). **d** Experimental timeline of the nerve injury and behavior tests in mice. **e** Compared with the sham group, the mechanical threshold of the ipsilateral hind paw with injury was significantly reduced in the nerve injury group (Sham, $n = 8$ mice; Nerve injury, $n = 12$, for Sham versus Nerve injury, $p < 0.001$ for 3 d, 7 d, 14 d, and 28 d). **f** Blue light evoked-activation of the ACC-ACC after nerve injury did not affect the withdrawal threshold of the ipsilateral hind paw ($n = 7$ mice). **g** Yellow light evoked-inhibition of the ACC-ACC significantly increased the withdrawal threshold of the ipsilateral hind paw after nerve-injured in mice ($n = 5$ mice, for Nerve injury versus Nerve injury + eNpHR, $p < 0.01$ for 3 d, $p < 0.05$ for 7 d, $p < 0.01$ for 14 d, $p < 0.001$ for 28 d). **h** Compared with the sham group, the mechanical threshold of the contralateral hind paw with injury was significantly reduced in the nerve injury group (Sham, $n = 8$ mice; Nerve injury, $n = 12$ mice, for Sham versus Nerve injury, $p < 0.01$ for 3 d and 28 d, $p < 0.001$ for 7 d and 14 d). **i** Blue light activation of ACC-ACC significantly decreased contralateral hind paw withdrawal threshold 3, and 14 days after nerve-injured in mice ($n = 7$ mice, for Nerve injury versus Nerve injury +ChR2, $p < 0.05$ for 3 d, $p < 0.01$ for 14 d). **j** Yellow light inhibition of ACC-ACC significantly increased the withdrawal threshold of the contralateral hind paw 7 days after nerve-injured in mice ($n = 5$ mice, $p < 0.01$ for Nerve injury versus Nerve injury +eNpHR of 7 d). Error bars represent SEM, *indicates significant difference (*$p < 0.05$, **$p < 0.01$, ***$p < 0.001$). ACC anterior cingulate cortex, BL baseline.

sample was embedded with an equal volume mixture of 4% HMS and 20% bovine serum albumin (BSA) at 37 °C for 4 h. The embedded block containing the sample was vertically glued to the base of the vibroslicer (Compresstome VF-300, Precisionary

Instruments). Then the brain was cut into 300-μm-thickness coronal slices. Brain slices were then treated following the two-step PuClear clearing method to obtain uniform optical transparency. Cleared brain slices were imaged at

**Table 1 Summary of studies on contralateral projections of brain regions.**

| Brain regions | Species | Contralateral projections | References |
|---|---|---|---|
| Cingulate cortex | Rats, mice | The first axons to cross the midline of the corpus callosum arise from the cingulate cortex. The ACC-contralateral ACC contributes to contralateral hindpaw pain. | 19,53 |
| RSG | Rats | Axons from the contralateral RSG form a dense terminal plexus in layers IV and V, with a smaller number of terminals in layers I and VI. | 54 |
| AI | Hamsters | The AI sends projections to layers I, V, and VI of the contralateral AI via the corpus callosum. | 55 |
| mPFC | Rats | Cells innervating the contralateral prefrontal cortex were distributed in layers II-III and V. | 56 |
| S1/S2 | Cats, rats | Callosal neurons which in the forepaw and hindlimb representations of S2 and the trunk representation of S1 or S2 projecting to the contralateral hemisphere are found in the cat. These callosal neurons are mainly located in layer III. Layer V neurons in the forelimb and shoulder representations in rat S1 project to layers II–VI in contralateral S1. | 57,58 |
| M1 | Rats | The M1 whisker or forepaw regions revealed robust projections to the contralateral corresponding M1 cortical areas. These presumably mediate bilateral coordination of the whiskers and limb. | 59 |
| Visual cortex | Rats | The visual callosal projection arises from pyramidal cells in layers II to VI b. Nonpyramidal neurons in layers II-III and V were found to project contralaterally. | 60 |
| Auditory cortex | Monkeys | Rostral (R) field→contralateral rostromedial (RM) and primary (A1) field; A1→contralateral RM and R; axon terminations are concentrated in layer IV of anterolateral (AL) and RM fields and upper layer III and layer IV of R and CM fields. | 61 |
| Olfactory bulb | Fogs | The olfactory bulb sends a projection to the medial part of the contralateral olfactory bulb. | 62 |
| MT | Galagos (primates) | Contralateral cortical terminations were in locations corresponding to the injection site of MT. | 63 |
| Amygdala | Pigeons | A small area of arcopallial and amygdaloid neurons constitute a wide range of contralateral projections to sensorimotor and limbic structures in birds. | 64 |
| Gi | Rats | The Gi of the medullary reticular formation (MRF) sends a projection to the contralateral Gi. | 65 |
| DMN | Rats | The DMN of midbrain sends a small number of projections to the contralateral DMN. | 66 |

*ACC* anterior cingulate cortex, *RSG* retrosplenial granular a cortex, *AI* agranular insular cortex, *mPFC* medial prefrontal cortex, *S1* primary somatosensory cortex, *S2* second somatosensory cortex, *M1* primary motor cortex, *MT* middle temporal area, *Gi* gigantocellular reticular nucleus, *DMN* deep mesencephalic nucleus.

$1 \times 1 \times 2.5 \, \mu m^3$ voxel resolution using the VISoR2 technique as described previously. Synchronized beam-scan illumination and camera-frame readout generated a stack of image frames of the 45° oblique optical sections of the sample with the sample stage moving linearly in the X-direction. Image volumes of each mouse brain slice were stitched with the custom software to automatically reconstruct the whole brain as described previously[48].

**Brain slice preparation**. Coronal brain slices (300 μm) containing the bilateral ACC were prepared using our previous methods[10,28]. Briefly, mice were anesthetized with 2% isoflurane and sacrificed by decapitation. The whole brain was rapidly removed from the skull and transferred into ice-cold oxygenated (95% $O_2$ and 5% $CO_2$) ACSF containing (in mM: 124 NaCl, 2.5 KCl, 2 $CaCl_2$, 2 $MgSO_4$, 25 $NaHCO_3$, 1 $NaH_2PO_4$, and 10 glucose, pH 7.3–7.4). For the making of coronal brain slices, the brain was trimmed and glued to the cutting staged tissue slicer (Leica, VT1200S). Then 300 μm-thickness slices containing the bilateral ACC were cut and transferred to a submerged recovery chamber with oxygenated (95% $O_2$ and 5% $CO_2$) ACSF at room temperature for at least 1 h.

**In vitro whole-cell patch-clamp recording**. The whole-cell patch recordings were performed as previously described[10,28,50]. Experiments were performed in a recording chamber placed in an Olympus BX51W1 microscope with infrared DIC optics for visualization of whole-cell patch clamp recording. The recordings were performed in voltage- or current-clamp mode using an Axon 200B amplifier (Molecular Devices). In the present study, AAV2/9-hSyn-hChR2(H134R)-EYFP-WPRE-hGH-pA was injected into the left ACC. The blue light was delivered onto the right ACC by fiber-coupled laser (200 mW, 465 nm, Inper Studio, Hangzhou), and evoked EPSCs (eEPSCs) were recorded in the left and right ACC. The recording pipettes (3–5 MΩ) were filled with a solution containing (in mM) 145 K-gluconate, 5 NaCl, 1 $MgCl_2$, 0.2 EGTA, 10 HEPES, 2 Mg-ATP, and 0.1 Na3-GTP (adjusted to

pH 7.2 with KOH, 290 mOsmol). Picrotoxin (100 μM) was always present to block $GABA_A$ receptor-mediated inhibitory synaptic currents in all experiments. To examine synaptic responses, the input (stimulus intensity)-output (EPSC amplitude) (I-O curves) relationships in the ACC pyramidal neurons were recorded at different stimulus intensities. Action potentials were recorded by delivering blue light stimulation at 5, 10, and 20 Hz in the left and right ACC neurons. The neurons were voltage-clamped at −60 mV in the presence of AP5 (50 μM) for AMPAR-EPSCs recordings. CNQX was added in ACSF to block selective non-NMDA ionotropic glutamate receptors. Access resistance was 15-30 MΩ and was monitored throughout the experiment. Data were discarded if access resistance changed by 15% during the experiment. Data were filtered at 1 kHz and digitized at 10 kHz.

**Two-photon calcium imaging**. The Two-photon calcium imaging was performed as previously described[26]. In vitro calcium imaging in the slice containing ACC was performed using a two-photon laser scanning microscope (Olympus FV1000-MPE system, BX61WI microscope) based on a pulsed Ti-sapphire laser (MaiTai HP DeepSee, 690–1040 nm wavelength, 2.5 W average power, 100 fs pulse width, 80 MHz repetition rate; New Port Spectra-Physics, Santa Clara, CA, USA). The laser was focused through a ×40 water-immersion objective lens (LUMPLFL/IR40XW, N.A.: 0.8, Olympus) and the average power was set to <15 mW (measured under the objective). Neurons were filled with indicators via the patch pipette for 20-30 min to allow diffusion of the dye into the cells. Fluorescent imaging of Cal-520 $K^+$ salt (200 μM) and Alexa594 $K^+$ salt (20 μM) was separated into green and red channels by a dichroic mirror and emission filters (Chroma, Bellows Falls, VT, USA), and detected by a pair of photomultiplier tubes (Hamamatsu, Shizuoka, Japan) at 800 nm. To obtain time series of fluorescent signals from global soma images, images were collected with the following parameters: 512 × 512 pixel images, digital zoom 3× with ×40 objective (N.A. 0.8), 2-μs pixel dwell time, 50 ms/frame for frame scan

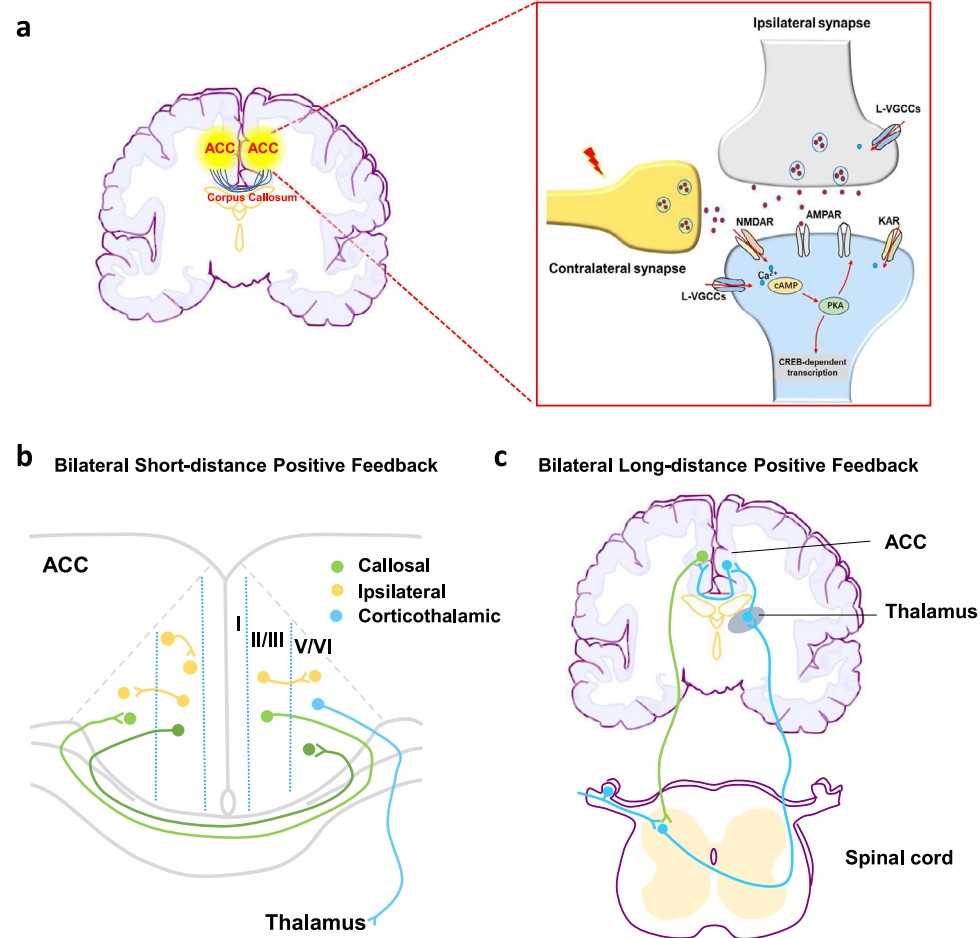

**Fig. 9 Signaling pathways and the projection circuit of the ACC-ACC. a** Signaling pathways that mediate the excitatory transmission of ACC to the contralateral ACC. Stimulation of ACC-ACC pre-synapses leads to AMPA/KA receptor-mediated postsynaptic excitatory response and an increase in calcium signaling. **b** Bilateral short-distance positive feedback projection in the internal ACC. Neurons from the superficial (II/III) and deep (V/VI) layers of the ACC have different output projections. Neurons located in layer II/III mainly project to the ipsilateral ACC and the contralateral ACC, while a mass of layer V/VI neurons project to the thalamus. **c** Bilateral long-distance positive feedback projection from the ACC. Peripheral nociceptive information is transmitted to the thalamus through the dorsal horn of the spinal cord. ACC neurons receive inputs from the thalamus. This study shows that ACC is directly projected to the contralateral ACC. Neurons in the deep layers of the ACC also send their projections directly or indirectly to the dorsal horn of the spinal cord. This positive spinal dorsal horn-thalamus-cortex-contralateral cortex-spinal dorsal horn loop may also provide a circuit for central sensitization. ACC anterior cingulate cortex.

model with different recording times for different recording frames. Bidirectional scanning and line-scanning models were used to increase scan speed. Each trial was repeated at least 3 times and the mean value was collected. Fluorescence changes were quantified as increases in green fluorescence from the baseline of ΔF/F = (F-F0)/F0 (Equation 1).

**Chronic neuropathic pain model**. Chronic neuropathic pain was induced through the ligation of the spared nerve injury (SNI) as described previously[51]. Mice were anesthetized using 2% isoflurane. Incisions were made on the skin and muscle of the left thigh to expose the sciatic nerve and its three branches. The tibial (TN) and the common peroneal nerves (CPN) were tightly ligated with 5-0 silk sutures and transected distal to the ligation sites. Leaving the sural nerve (SN) intact. The overlying muscle and skin were sutured. Sham surgery included exposure of the sciatic nerve, but all nerves were not injured. All animals were kept in a warming chamber for at least 30 min after surgery.

**Behavioral tests**. The mechanical withdrawal threshold was determined using the up-down method as previously reported[28].

Mice were individually placed into a plastic cage with wire mesh floors and allowed to acclimate for 30 min before testing. The von Frey filaments were applied perpendicularly to the plantar surface of the paw until it buckled slightly and was held for 3-6 s. Positive responses include licking, biting, and sudden withdrawal of the hind paw. A series of filaments have different bending forces, including 0.008, 0.02, 0.04, 0.16, 0.4, 0.6, 1, 1.4, and 2.0 g. An initial filament force of 0.4 g was applied to the mice. If a negative response occurred, the filament force was incrementally increased until a positive response was obtained. If the positive response occurred with 0.4 g filament force, the filament force was incrementally decreased until a negative result was obtained. This up-down method was repeated until five changes in behavior were determined. An interval of 3–5 min is required between every two positive results.

The hot plate test was performed on a hot plate at 55 ± 1 °C. The latency time in the first positive reaction of the hind paws was recorded. Positive responses include lifting, licking, shaking, and jumping. The cut-off time is 20 s to avoid tissue damage. The test was repeated three times with an interval of 10 min. The final latency time is the average of the three measurements.

The tail-flick test was measured using a 50 W projector lamp. The latency time to reflexive removal of the tail from radiant heat was recorded. The cut-off time is 10 s to avoid tissue damage. The test was repeated three times with an interval of 10 min. The final latency time is the average of the three measurements.

The open-field test was performed as previously described. Mice were placed in an open field (40 × 40 × 30.5 cm) and allowed to explore freely for 25 min with a 5 min light stimulation interval. Define the 20 × 20 cm in the center of the open field as the center zone and the rest as the periphery zone. A tracking master v3.0 system was used to record and analyze horizontal locomotor activity. Total distance, the number of center entries, and time spent in the center were recorded.

The elevated plus maze (EPM) test was performed as previously described. Its apparatus consisted of two open arms (30 × 5 cm) and two closed arms (30 × 5 × 30 cm). The open and closed arms are perpendicular to each other. For each test, the individual mouse was placed in the center of the apparatus and allowed to move freely for 5 min. Total distance, the number of entries, and time spent in each arm were recorded.

The rotarod test was performed as previously described. 1 h before the test, the mice were trained to stay on the rotating drum for 30 s at a constant acceleration of 16 rpm. When tested, the Rotarod will accelerate from 4 rpm to 40 rpm over a 5 min period. 5 min is set as the maximum time per session. Mice were tested 3 times at an interval of 5 min. The latency period of a fall is recorded as a measure of motor function.

The conditioned place avoidance (CPA) test was performed in a three-chamber apparatus (Med associates) consisting of two large chambers of the same size ($20 \times 20 \times 20\ cm^3$) and a middle corridor ($20 \times 10 \times 20\ cm^3$). The two large chambers have distinct walls drawings and floors. Animal movements were recorded using a tracking system (Shanghai Vanbi Intelligent Technology Co., Ltd.). According to a previous study, the paradigm consisted of 3 phases over 5 days. During the preconditioning phase on the first day, the mice were placed in the apparatus for 15 min and were allowed to move freely with access to all three chambers. Animals spending >50% or <30% of the total time in a big chamber were excluded from further testing. Over the following 3 d of the conditioning phase, the mice received blue light (465 nm, 2–5 mW, 20 Hz, 5 ms pulse) for 15 min in a random large chamber. Six hours later, the mice were connected to an optical fiber but the lasers were off for 15 min in the other large chamber. During the testing phase on the fifth day, the mice were allowed to move freely with access to all three chambers, and their movements were recorded for 15 min. CPA score is the result of deducting the time spent during the testing phase in the light-paired chamber minus the time spent during the preconditioning phase.

**Optical manipulations**. According to the above described, 200 nL of anterograde tracer virus AAV2/9-hSyn-hChR2 (H134R)-EYFP-WPRE-hGH-pA or AAV2/9-hSyn-eNpHR-mCherry-WPRE-hGH-pA was injected into the unilateral ACC. The optic fiber (length, 2 mm; 200-μm core; NA = 0.37; THINKERTECH, Nanjing) was implanted into the other side of ACC to activate or suppress ACC terminals. In addition, 200 nL of anterograde tracer virus AAV2/9-hSyn-hChR2 (H134R)-EYFP-WPRE-hGH-pA or AAV2/9-hSyn-eNpHR-mCherry-WPRE-hGH-pA was injected into the bilateral ACC. The optic fibers were implanted into the bilateral ACC.

Behavioral tests combined with optogenetics were performed 2 weeks after recovery from surgery. The optical fiber is connected to fiber-coupled lasers to control the frequency and intensity of the light. For EPM, and open field, blue light (465 nm, 2–5 mW, 20 Hz, 5 ms pulse) was delivered by three light 'off' and

two light 'on' sessions (off1-on1-off2-on2-off3, each 'off' or 'on' is 5 min)[27,52]. For mechanical withdrawal threshold, hot plate, and Rotarod tests, mice received blue or yellow light (593 nm, 5–8 mW, constant) illumination during testing. For the tail-flick test, mice received blue light illumination for 5 min. The latency time of the tail flick was measured several times between 10 min before and 1 h after light exposure. After the experiments, all mice were sacrificed and the whole brain was sectioned to examine the location of virus expression and optical fiber implantation. If virus expression or fiber implantation deviated from the target area, the data were excluded.

In vitro, whole-cell patch-clamp recording combined with optogenetics was performed 3 weeks after virus injection. The optic fiber was localized in the recording chamber at 3 mm from the recorded neurons of the target area. Blue light (465 nm, 15 mW, 5 and 10 Hz, 5 ms pulse) was delivered. The ACC neurons of the virus-injected side were recorded in a current-clamp at a holding current allowing maintaining the resting membrane potential, and the ACC neurons of the other side were fixed at −60 mV in voltage-clamp mode.

**Statistics and reproducibility**. All data were reported as the means ± standard error of the mean (SEM). Data were analyzed and plotted with OriginPro 8.0 or GraphPad Prism 8.0 software. For the whole-cell patch-clamp recording, Clampex and Clampfit 10.2 software were used to acquire and analyze the data. For comparison between the two groups, statistical significance was assessed using unpaired Student's $t$-test or paired t-test. For comparison among three or more groups, statistical significance was assessed using one-way analysis of variance ANOVA or two-way ANOVA (Student-Newmann-Keuls or Tukey test was used for post-hoc comparisons). In all cases, $p < 0.05$ was considered statistically significant.

**Reporting summary**. Further information on research design is available in the Nature Portfolio Reporting Summary linked to this article.

### Data availability
Source data underlying figures are provided in Supplementary Data 1. Other datasets generated and analyzed during the current study are available from the corresponding author upon reasonable request.

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

## Acknowledgements

The authors thank Melissa Lepp for English editing. We would like to thank the EJLB-CIHR Michael Smith Chair in Neurosciences and Mental Health in Canada, Canada Research Chair, Ontario-China Research and Innovation Fund (OCRIF), Canadian Institute for Health Research operating and project Grants (MOP-124807; PJT-148648 and 419286) for funding support to M.Z. X.-H.L. was supported by grants from the National Natural Science Foundation of China (32100810) and the Fundamental Research Funds for the Central Universities (xxj032022013 and xzy012022046).

## Author contributions

X.H.L. and M.Z. designed the project. X.H.L., W.S., and M.Z. drafted the manuscript and finished the final version of the manuscript. X.H.L. and W.S. performed the optogenetic and behavior experiments. X.H.L., Q.Y.C., S.H., M.H.H., and Z.M. performed the electrophysiological experiments. X.H.L. and S.H. performed the two-photon calcium imaging. W.S., F.X., and G.Q.B. performed the immunohistochemical and morphological experiments. All authors read and approved the final manuscript.

## Competing interests

The authors declare no competing interests.
