## [Peer Review File · Communications Biology]

Reviewers' comments:

Reviewer #1 (Remarks to the Author):

The manuscript by Li et al. addresses a long-standing question in neuroscience, namely about the functional role of bilateral connectivity in some cortical areas across the two hemispheres of the brain. Here, using an elegant combination of slice physiology, in vivo optogenetics and behavioural analyses, the authors test the role of inter-connectivity of cingulate cortical areas in nociceptive hypersensitivity and social interactions in a model of chronic pain in mice. The manuscript thus adds onto recent studies testing this question and importantly, provides detailed electrophysiological analyses on the nature of this contralateral connectivity and its activity-dependent plasticity.

The study is very well-performed and adds important and novel insights on a long-standing question. This reviewer only has a few minor comments:

Main points:

1. The anterior cingulate cortex (ACC) is represented as a single, homogenous area in the introduction and the results. Rather, it comprises several functionally-heterogenous domains, that play different roles in pain and related emotions (e.g., see Tan et al. Nature Neuroscience 2017, several reviews by BA Vogt). Therefore, the authors should clearly represent in the text and figures which area of the ACC is being targeted in this study.
2. Do the excitatory bilateral ACC connections described on pages 4 and 5 in electrophysiological analyses only exist between pyramidal neurons, or do local interneurons (GABAergic) also receive these connections? If this was not tested experimentally, this point should at least be discussed.
3. Line 166: The pure AMPA-mediated connectivity found in some pyramidal neurons is interesting. Was the converse (i.e. pure NMDA synapse, so-called 'silent synapse') also seen in any cells?
4. Paragraph starting on line 170: it does not become clear whether the two photon imaging experiments described for recording calcium transients performed on slices or in vivo. Please describe clearly.
5. Lines 201-208: Anxiety was only evident upon optogenetic stimulation of bilateral ACC connectivity with respect to one of the parameters in the EPM test and was not seen in the open field test. Therefore, it would be appropriate to tone down the strong claim that bilateral ACC connectivity induces anxiety (line 208, abstract and discussion), since this was not a really consistent observation.
6. Line 218: 'Similarly, optic-activation of the left ACC decreased the mechanical withdrawal thresholds of the right hind paw, and had no effect on the nociceptive response of the right hind paw (Fig. 6B)'. This statement does not make sense logically since a change in mechanical thresholds is obviously a part of the nociceptive response.
7. Line 226: The observation of increase in nociceptive thresholds upon ACC inhibition is contrary to all previous reports from other groups on analysis of ACC in pain and would implicate that the ACC exerts a tonic facilitation of pain. How is this to be explained? Is the ACC tonically active? Previous studies do not show that, rather it is demonstrated to be activated in response to a noxious stimulus.
8. Line 236: The authors report contralateral hypersensitivity in the unaffected limb in the SNI model. This diverges from the initial description of the model and other studies. Has mirror pain been reported in this model previously? Also, it is difficult to call mechanical hypersensitivity as 'mirror pain' since pain implies negative affect. It would be better to tone down this aspect unless negative affect is tested transparently and clearly, e.g., using conditioned place preference or other tests.

9. Only male mice were tested in this study, thus making it unclear whether the results hold true in both sexes. This needs to be discussed.

9. Thorough editing is required throughout the text. Some examples: Line 83: 'consisted' should be changed to 'consists'; Line 108: 'right of the ACC' should be changed to 'ACC in the right hemisphere of the brain' or similar.

Reviewer #2 (Remarks to the Author):

In this study, Li et al. investigated direct cortico-cortical projections between left and right ACC regions. Using a combination of anatomical tracing, in vitro EP, in vivo 2-photon imaging, and behavior testing in the context of optogenetics and in vivo pharmacology, the authors show that ACC-ACC connectivity facilitates behavioral withdrawal responses to noxious mechanical and thermal stimuli and enhanced anxiety-like behaviors associated with acute pain. In addition, they show that this connection also plays a role in chronic pain, using the neuropathic pain (SNI) model. The study is innovative and provides important insight into cortical pain mechanisms, and experiments are performed with technical sophistication. I have the following comments.

1. While the authors have conducted a number of behavioral assays to show the importance of this pathway in pain processing, the ACC is known to be important for pain aversive processing. It would be interesting to see if ACC-ACC connectivity could affect pain aversion. The authors should at least comment on this role, especially in the context of a large body of literature in rodent models.
2. Activation of axon terminals with ChR2 carries a potential confound of retrograde action potential activation, especially in behavioral assays. The authors may want to discuss this possibility. This is less of an issue with NpHR, however.
3. Bidirectional ACC-ACC connectivity raises an interesting question of laterality for nociceptive processing. The authors may want to elaborate further on this point. This is especially salient for pain-aversive processing that is thought to be less anatomically defined and may not display overt laterality.

Reviewer #1

1. The anterior cingulate cortex (ACC) is represented as a single, homogenous area in the introduction and the results. Rather, it comprises several functionally-heterogenous domains, that play different roles in pain and related emotions (e.g., see Tan et al. Nature Neuroscience 2017, several reviews by BA Vogt). Therefore, the authors should represent in the text and figures which area of the ACC is being targeted in this study.

Answer: Thanks for the helpful suggestion. The mouse ACC covering Brodmann areas 24a/24b were targeted in our study. We have provided additional information in the text and figures of the revised manuscript.

2. Do the excitatory bilateral ACC connections described on pages 4 and 5 in electrophysiological analyses only exist between pyramidal neurons, or do local interneurons (GABAergic) also receive these connections? If this was not tested experimentally, this point should at least be discussed.

Answer: Thanks for the helpful suggestion. In our study, we did not test the excitatory projections between pyramidal neurons to interneurons. In agreement with this reviewer's suggestion, there are reports of inhibitory callosal projection neurons in the cortex, although most callosal projection neurons are excitatory neurons (Zurita et al., 2018). In our study, we did find the retrograde neuron labeled by rabies virus in layer I of contralateral ACC (Figs. 1D-1F) and anterograde fibers labeled by AAV2/9 virus in layer I of contralateral ACC (Figs. 1G-1I). These results indicate that there are inhibitory projections between interneurons to pyramidal neurons in bilateral ACC or excitatory projections between pyramidal neurons to interneurons. We have added more discussion in the revised manuscript, and future studies will be needed to reveal inhibitory projections in bilateral ACC.

3. Line 166: The pure AMPA-mediated connectivity found in some pyramidal neurons is interesting. Was the converse (i.e. pure NMDA synapse, so-called 'silent synapse') also seen in any cells?

Answer: In our previous research, we found that there are pure NMDA synapses (also called 'silent synapses') in the ACC, although we have not detected the pure NMDA synapses in the ACC-ACC synapse in the present study. Together with our current results, we found that cortical synapses in the ACC are likely heterogeneous. There are at least three different types of excitatory synapses: silent synapses; AMPA receptor-containing synapses, and AMPA and kainate mixed synapses. While we still need direct evidence for silent synapses in the ACC, our previous field recording electrophysiology data consistently indicates the existence of some silent 'responses', or pure NMDA receptor-mediated responses in the ACC, and the recruitment of such silent synapses by LTP or chemicals (Li et al., Mol. Pain 2019; Song et al., J Neurosci. 2017; Chen et al., Mol Brain 2014). In a previous study of freely moving animals, Wu et al. (2005) reported that ACC stimulation induced mixed fast and slow responses in the contralateral ACC (Wu et al., J Neurosci. 2005). These findings indirectly suggest the possible existence of pure NMDA synapses in the ACC.

4. Paragraph starting on line 170: it does not become clear whether the two-photon imaging experiments described for recording calcium transients were performed on slices or in vivo. Please describe clearly.

Answer: Thanks for your suggestion. The two-photon imaging experiments were all performed on slices in the ACC. We have clarified the description of the results and methods in the revised manuscript.

5. Lines 201-208: Anxiety was only evident upon optogenetic stimulation of bilateral ACC connectivity with respect to one of the parameters in the EPM test and was not seen in the open field test. Therefore, it would be appropriate to tone down the strong claim that bilateral ACC connectivity induces anxiety (line 208, abstract and discussion) since this was not a consistent observation.

Answer: Thanks for your good suggestion. We have toned down the claim that bilateral ACC connectivity induces anxiety in the revised manuscript. The previous claim has been deleted in the abstract and weakened in the results and discussion. Future studies are needed to investigate this.

6. Line 218: ‘Similarly, optic-activation of the left ACC decreased the mechanical withdrawal thresholds of the right hind paw, and did not affect the nociceptive response of the right hind paw (Fig. 6B)’. This statement does not make sense logically since a change in mechanical thresholds is a part of the nociceptive response.

Answer: Thanks for the suggestions. We made corrections to this statement.

7. Line 226: The observation of an increase in nociceptive thresholds upon ACC inhibition is contrary to all previous reports from other groups on analysis of ACC in pain and would implicate that the ACC exerts a tonic facilitation of pain. How is this to be explained? Is the ACC tonically active? Previous studies do not show that, rather it is demonstrated to be activated in response to a noxious stimulus.

Answer: Sorry for the confusion. For this ACC inhibition experiments performed in animals with nerve injury, ACC and ACC-induced top-down facilitation are believed to be activated (see Chen et al., 2018). Since ACC projections are bilaterally on spinal nociceptive transmission, it is conceivable that contralateral nociceptive responses were also affected. We believe that this top-down facilitation modulation is tonically active after peripheral injury as the reviewer pointed out.

8. Line 236: The authors report contralateral hypersensitivity in the unaffected limb in the SNI model. This diverges from the initial description of the model and other studies. Has mirror pain been reported in this model previously? Also, it is difficult to call mechanical hypersensitivity as ‘mirror pain’ since pain implies a negative effect. It would be better to tone down this aspect unless the negative affect is tested transparently and, e.g., using conditioned place preference or other tests.

Answer: Thank you for your helpful suggestion. We have revised the text and removed the ‘mirror’ pain in the revised manuscript.

9. Only male mice were tested in this study, thus making it unclear whether the results hold in both sexes. This needs to be discussed.

Answer: Thanks for the helpful suggestion. We have decided to perform additional experiments using female mice and new data were included in the revised manuscript. Basically, we found that the ACC-ACC effect also exists in female animals.

10. Thorough editing is required throughout the text. Some examples: Line 83: ‘consisted’ should be changed to ‘consists’; Line 108: ‘right of the ACC’ should be changed to ‘ACC in the right hemisphere of the brain’ or similar.

Answer: Thanks for the suggestion, and we have revised the text accordingly.

Reviewer #2

1. While the authors have conducted several behavioral assays to show the importance of this pathway in pain processing, the ACC is known to be important for pain-aversive processing. It would be interesting to see if ACC-ACC connectivity could affect pain aversion. The authors should at least comment on this role, especially in the context of a large body of literature on rodent models.

Answer: First we like to thank this reviewer for his/her appreciation of our work. As suggested, the aversive behaviors were tested both in the male and female mice by light-activating the ACC-ACC pathway. We found that light-activating ACC-ACC connectivity does not affect aversive behaviors in the physiology condition both in male and female mice (Fig. 5 and Fig. 6). However, light-activating ACC-ACC connectivity enhanced aversive behaviors in SNI model mice (see Results).

2. Activation of axon terminals with ChR2 carries a potential confound of retrograde action potential activation, especially in behavioral assays. The authors may want to discuss this possibility. This is less of an issue with NpHR, however.

Answer: Thanks for a good suggestion. We agree that ChR2 may carry a potential confound of retrograde action potential activation in the behavioral setting. We have now added the discussion in the revised manuscript.

3. Bidirectional ACC-ACC connectivity raises an interesting question of laterality for nociceptive processing. The authors may want to elaborate further on this point. This is especially salient for pain-aversive processing that is thought to be less anatomically defined and may not display overt laterality.

Answer: Thanks for this great suggestion. As with comment 1, in the revised paper, the aversive behaviors were tested in the male and female mice by light-activating ACC-ACC connectivity. We found that light-activating ACC-ACC connectivity does not affect aversive behaviors in the physiology condition both in male and female mice (Fig. 5 and Fig. 6). However, light-activating ACC-ACC connectivity can enhance aversive behaviors in nerve injury model mice. As suggested by this reviewer, we have now added additional discussions regarding somatosensory pain processing vs pain aversive process in the revised Discussion.

REVIEWERS' COMMENTS:

Reviewer #1 (Remarks to the Author):

The authors have done a very good job in revising the manuscript. All of my comments were addressed well. This is an interesting and important study.

Reviewer #2 (Remarks to the Author):

The authors have done a good job addressing my concerns. This is an interesting study.

Reviewer #1

The authors have done a very good job in revising the manuscript. All of my comments were addressed well. This is an interesting and important study.

Answer: We like to thank this reviewer for his/her appreciation of our work. Thanks again for taking the valuable time to review our paper.

Reviewer #2

The authors have done a good job addressing my concerns. This is an interesting study.

Answer: We like to thank this reviewer for his/her appreciation of our work. Thanks again for taking the valuable time to review our paper.